# Don't Miss the Forest for the Trees: Attentional Vision Calibration for Large Vision Language Models

## Abstract

This study seeks to understand and address a phenomenon observed in Large Vision Language Models (LVLMs) related to their attention mechanism. Interestingly, LVLMs tend to disproportionately focus on a few image tokens that lack meaningful, query-related semantics, leading to sharp outlier values in the attention maps — tokens we refer to as *blind tokens*. In well-designed attention mechanisms, the principle is to assign higher weights to the most relevant tokens. However, in this case, the attention imbalance leads to overemphasis on uninformative tokens, which is far from ideal. Our analysis shows that tokens receiving lower attention weights often hold critical information necessary for capturing subtle visual details. We hypothesize that over-reliance on blind tokens contributes to hallucinations in LVLMs. To address this, we introduce a novel decoding technique called Attentional Vision Calibration (AVISC). During the decoding phase, AVISC identifies blind tokens by examining the image-wise attention distribution and dynamically adjusts the logits for the prediction. Specifically, it contrasts the logits conditioned on the original visual tokens with those conditioned on the blind tokens, thereby reducing the model's dependency on blind tokens and encouraging a more balanced consideration of all visual tokens. We validate AVISC on standard hallucination benchmarks, including POPE, MME, and AMBER, where it consistently outperforms existing decoding techniques.

## 1 Introduction

Large Vision Language Models (LVLMs) (Dai et al., 2024; Zhu et al., 2023; Liu et al., 2023c;b; Bai et al., 2023; Tong et al., 2024a) have demonstrated remarkable capabilities in generating coherent and contextually relevant descriptions from visual inputs. This success largely hinges on the models' ability to interpret and integrate complex visual information with textual data. However, a persistent challenge with these models is their tendency towards "hallucinations" — producing inaccurate or fabricated descriptions that do not accurately reflect the visual data. The phenomenon of hallucination in LVLMs can significantly impede their reliability, especially in applications requiring precise and trustworthy visual descriptions.

Modern LVLMs (Dai et al., 2024; Liu et al., 2023c) are predominantly based on transformer architecture (Vaswani et al., 2017), where the most critical component is the attention mechanism. In this framework, the highest attention weights are assigned to tokens that the model considers most important for generating the output. This concept implies that tokens with higher attention are essential ingredients in the generation process, directly guiding the model's output choices.

The intuitive alignment between attention weights and key tokens is a well-established principle. For instance, the DINO (Caron et al., 2021) or OpenCLIP Ilharco et al. (2021) produce attention maps that naturally concentrate on semantically meaningful regions of an image. Textual attention mechanisms (Vaswani et al., 2017) were initially developed with a similar intuitive basis. It is worth to question whether these attention mechanisms in LVLMs truly align with their intended purpose. Our investigation reveals that LVLMs (Liu et al., 2023c; Dai et al., 2024) exhibit biased attention toward specific image tokens, which we refer to as "*blind tokens*". These tokens, despite receiving high attention, are not crucial for prediction or semantic understanding.

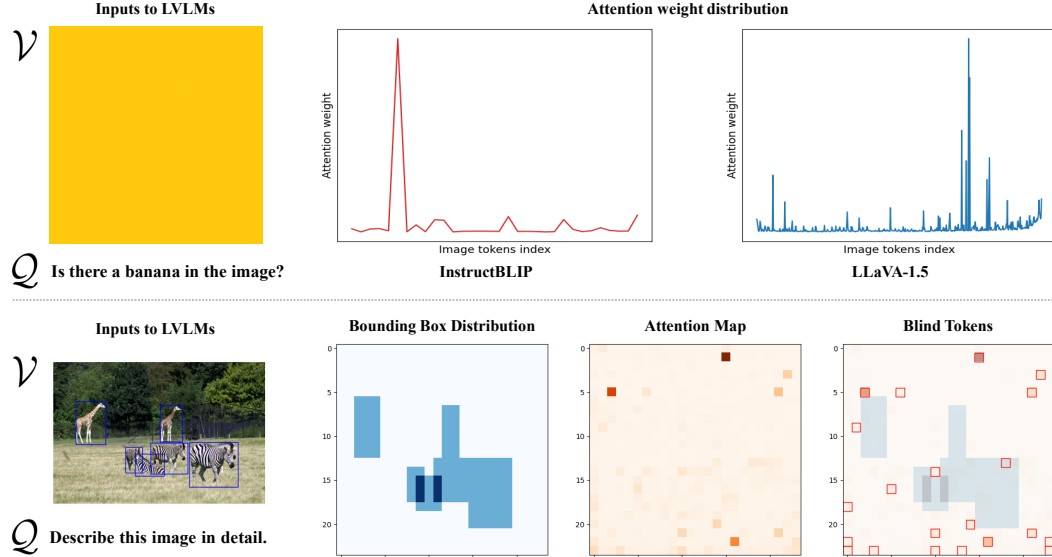

Figure 1: **Blind tokens in LVLMs. (Top)** Even when the image ($\mathcal{V}$) lacks information relevant to the textual query ($\mathcal{Q}$), LVLMs (Dai et al., 2024; Liu et al., 2023c) tend to to focus disproportionately on a few image tokens (*i.e.*, *blind tokens*). This phenomenon is observed by averaging attention weights across all layers when generating the first response. **(Bottom)** An overlay of bounding boxes and the attention map of LLaVA 1.5 highlights this effect, with blind tokens marked by red boxes. More examples are shown in Appendices B.1 and B.4.

This phenomenon aligns with the findings in (Darcet et al., 2023). They identified artifacts in the feature maps of vision transformers (Touvron et al., 2022; Caron et al., 2021; Oquab et al., 2023). These artifacts primarily appear in low-informative background areas of images during inference, where such regions receive disproportionately high attention despite containing minimal local information. (Darcet et al., 2023) suggest that, during attention operations, these tokens are repurposed to aggregate global image information while discarding spatial details, likely due to their association with repetitive or less informative image patches.

We are curious whether these counterintuitive attentional biases may be inherent to attention mechanisms themselves. Our work seeks to investigate whether this phenomenon extends beyond Vision Transformers to LVLMs and how differences in architecture, task, and domain contribute to attentional behavior. Building on (Darcet et al., 2023), we aim to explore how these attentional patterns manifest in LVLMs.

This issue becomes particularly apparent in image with uniform pixel values, where no specific region of the image warrants special attention. As illustrated in Fig. 1 (Top), even when an image contains no objects and consists only of a uniform yellow background, some regions still receive disproportionate focus. These blind tokens highlight potential flaws in how the model interprets visual data during the decoding process, as LVLMs (Dai et al., 2024; Liu et al., 2023c) often focus on irrelevant tokens.

We analyze the correlation between actual object regions and attention weights in LVLMs using the COCO2014 dataset Lin et al. (2014). We ask LVLMs to describe the images and analyzed attention distribution across $24 \times 24$ patches, comparing bounding boxes to attention weights on these patches. An example is shown in Fig. 1 (Bottom). Specifically, we assess the proportion of blind tokens within the bounding boxes. The results show that, on average, only 3.7% of actual object regions overlap with blind tokens, and only 23.3% of attention weights are assigned to the object regions, while the rest focused on other areas. This suggests that, the model's actual focus does not well align with object regions, which are crucial for accurate image descriptions.

We further examine the attention distribution of LLaVA-1.5 (Liu et al., 2023c) in response to the given image and textual query. A closer look at the functional impact of attention weights on the model's responses shows interesting insights (see Fig. 2): zeroing out blind tokens – those receiving excessive attention – has little effect on the original prediction logits, suggesting that LVLMs often assign high attention to tokens that lack object-discriminative information. In contrast, zeroing out non-blind

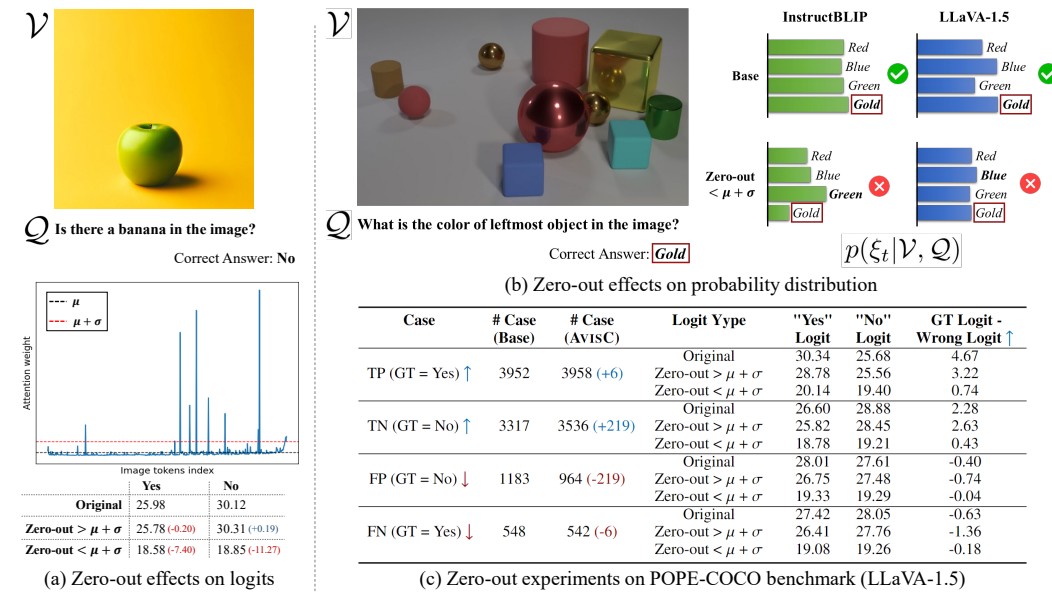

(b) Zero-out effects on probability distribution

(a) Zero-out effects on logits

(c) Zero-out experiments on POPE-COCO benchmark (LLaVA-1.5)

Figure 2: **Measuring impact of blind/non-blind tokens with zero-out experiments. (a)** Zeroing out image tokens with attention weights above $\mu + \sigma$ (mean + standard deviation), *i.e.*, *blind tokens*, does not significantly affect the original logits. This suggests that LVLMs assign high attention to tokens that lack object-discriminative information. Conversely, zeroing out *non-blind tokens* drastically disrupts the logits, often leading to near 50:50 probabilities, indicating a significant loss of discriminative information. **(b)** When non-blind tokens are zeroed out, the models fails to correctly predict previously well-classified instances. **(c)** Across the POPE-COCO benchmark using the LLaVA-1.5 model, zeroing out blind tokens (Zero-out $> \mu + \sigma$) has a smaller impact on prediction logits than zeroing out non-blind tokens (Zero-out $< \mu + \sigma$). AVISC effectively reduces over-emphasis on blind tokens, improving performance, especially in TN and FP cases.

tokens drastically alters the prediction logits, resulting in near-equal probabilities, indicating the loss of critical object-discriminative information. These highlight the need for a more balanced consideration of the entire image.

Such skewed attention, which disproportionately favors blind tokens while overlooking non-blind tokens containing crucial visual details, can lead to misclassifications or entirely incorrect predictions. We hypothesize that this over-reliance on blind tokens contributes to hallucinations in LVLMs.

In response to this challenge, we propose a novel method termed Attentional Vision Calibration (AVISC), which recalibrates the model's attention during the decoding phase. Unlike existing approaches that typically require extensive training (Jiang et al., 2023; Sun et al., 2023; Zhou et al., 2023; Yu et al., 2023b) or auxiliary models (Zhao et al., 2024; Wan et al., 2024; Deng et al., 2024; Yang et al., 2024; Li et al., 2023b), AVISC operates without these prerequisites.

AVISC dynamically modifies the decoding process in three steps: **(i)** Based on our findings that different LVLMs and across layers exhibit different attentional patterns (see Fig. 4), we first select relevant layers that allocate a higher proportion of attention to image tokens. **(ii)** Next, we identify *blind tokens*, which disproportionately dominate attention. These tokens are isolated, while all other image tokens are zeroed out, creating a biased visual input. **(iii)** Finally, we employ a contrastive decoding (Leng et al., 2023; Favero et al., 2024). This technique contrasts the logits derived from the original visual input with those derived solely from the blind tokens. By doing so, it amplifies the influence of tokens that exhibit significant differences between the two distributions.

Notably, AVISC does not directly manipulate attention weights. Instead, it adjusts the influence of blind and non-blind tokens at the prediction level using contrastive decoding, reducing the impact of blind tokens while enhancing the influence of non-blind ones.

Through a series of experiments on hallucination benchmarks like POPE (Rohrbach et al., 2018), MME (Fu et al., 2024), and AMBER (Wang et al., 2023b), we demonstrate that AVISC significantly mitigates hallucination while simultaneously improving the models' ability to capture and describe detailed image attributes more accurately.

## 2 RELATED WORK

LVLMs (Li et al., 2023a; Zhu et al., 2023; Chen et al., 2023a) are prone to generating hallucinations, *i.e.*, misalignment between visual inputs and textual outputs. These hallucinations manifest across various semantic dimensions such as incorrect object presence, attributes, or relations.

To mitigate these, researchers have developed strategies across three levels:

**Input-level.** Efforts here focus on data quality improvement to reduce hallucinations (Gunjal et al., 2023; Liu et al., 2023a; Wang et al., 2023a; Lu et al., 2024), including the introduction of negative data (Liu et al., 2023a), counterfactual data (Yu et al., 2023a) to challenge the model's assumptions, dataset cleansing to minimize noise and errors (Wang et al., 2024; Yue et al., 2024).

**Model-level.** This includes increasing the resolution at which models process visual data (Chen et al., 2023b; Liu et al., 2023b; Zhai et al., 2023), or enhancing perception abilities through advanced vision encoders (He et al., 2024; Jain et al., 2023; Tong et al., 2024b). These are usually training-based (Jiang et al., 2023; Yue et al., 2024), and often involve auxiliary supervision from external datasets (Chen et al., 2023c) and reinforcement learning techniques (Zhao et al., 2023; Gunjal et al., 2024; Sun et al., 2023; Yu et al., 2023b) to better align model outputs with accurate visual representations.

**Output-level.** Techniques like contrastive decoding (Leng et al., 2023; Favero et al., 2024) directly contrast incorrect predictions during decoding, helping models distinguish between accurate and inaccurate descriptions. Guided decoding (Zhao et al., 2024; Deng et al., 2024; Chen et al., 2024) leverages external models like CLIP (Radford et al., 2021) or DETR (Carion et al., 2020) to enhance accuracy. Other approaches include training-free methods (Wan et al., 2024; Zhang et al., 2024; Huang et al., 2023) and post-hoc corrections via self-feedback (Lee et al., 2023; Wu et al., 2024).

Among these, we focus on contrastive decoding methods: (1) VCD (Leng et al., 2023) mitigates statistical biases and language priors by contrasting output distributions from original and distorted visual inputs, moderating decoding probabilities. (2) M3ID (Favero et al., 2024) uses a similar approach where the reference image amplifies its influence over the language prior, thereby enhancing the generation of tokens with higher mutual information with the visual prompt.

Our approach belongs to the output-level category. AVISC analyzes attention patterns to identify *blind tokens* during decoding steps. It then utilizes a contrastive decoding technique to enhance token prediction. Our method does not require additional training, external data or models, and costly self-feedback mechanisms.

## 3 APPROACH: AVISC

We propose a straightforward method, called AVISC, to enhance visual object understanding in LVLMs during the decoding phase. AVISC dynamically calibrates the over-emphasis on *blind tokens* on-the-fly at every token generation step. The calibration is guided by the attention patterns of image tokens in response to the given image and textual query. Importantly, AVISC operates without additional training, external models, or complex self-feedback mechanisms. A visual summary of our method is shown in Fig. 3. AVISC modifies the

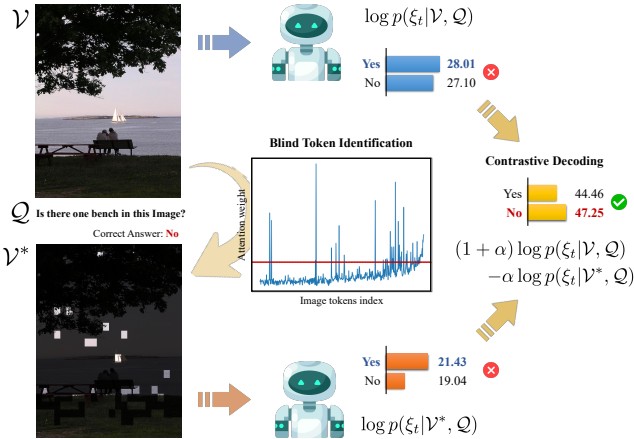

Figure 3: **An overview of AVISC.**

decoding process in three steps: **(1) Layer selection**: choose layers significantly influenced by image tokens, **(2) Blind token identification**: detect non-relevant tokens in selected layers, and **(3) Contrastive decoding**: adjust the decoding process to mitigate the influence of blind tokens.

### 3.1 LVLM FRAMEWORK

**Uni-modal encoding.** LVLM begins by encoding visual inputs and textual queries into compact representations. Visual inputs provide contextual information that helps generate responses relevant

to the textual queries. The text data is tokenized, turning it into a sequence of manageable pieces for further processing. For visual data, a commonly used encoder is a pre-trained model like CLIP (Radford et al., 2021), which is already semantically aligned with textual data through extensive training on image-text pairs.

**Cross-modal alignment.** As LLM inherently perceives only text, aligning text and vision modalities is essential. Instead of retraining LLM, which would be prohibitively expensive, a more viable approach is to use a learnable cross-modal alignment module. This module, such as Q-Former (Li et al., 2023a) or a linear projection layer (Liu et al., 2023c), transforms visual features into a format compatible with the LLM's input space. This process results in a set of visual tokens, $\mathcal{V} = \{v_0, v_1, \ldots, v_{N-1}\}$, which are then concatenated with the text tokens, $Q = \{\sigma_N, \sigma_{N+1}, \ldots, \sigma_{N+M-1}\}$, to form a unified input sequence of length $N + M$.

**Next token prediction via LLM.** The concatenated sequence of visual and textual tokens is then processed by LVLM, parametrized by $\theta$, which generates responses in an auto-regressive manner. The model calculates logits for each potential next token:

$$\ell_t = \log p(\xi_t | \mathcal{V}, Q, \xi_{<t}; \theta), \tag{1}$$

where $\ell_t$ are the logits for the next token at timestep $t$, $\xi_t$ denotes the next token being predicted, and $\xi_{<t}$ represents the sequence of tokens generated up to timestep $(t - 1)$. From these logits, we apply a softmax function to convert logits into a normalized probability distribution:

$$p(\xi_t) = \text{Softmax}(\ell_t). \tag{2}$$

The next token $\xi_t$ is sampled from this probability distribution, with the model continuing this predictive process until the response sequence is complete.

### 3.2 ATTENTIONAL VISION CALIBRATION FOR ALLEVIATING HALLUCINATIONS

Visual hallucinations in LVLMs can emerge during the decoding phase when the model selects tokens based on erroneous probability distributions that do not align with the visual inputs. These discrepancies, as demonstrated in our observations (refer to Figs. 1 and 2), often originate from an attentional bias toward certain non-relevant tokens, referred to as *blink tokens*. Our methodology aims to recalibrate these attention patterns to correct such hallucinations.

**Layer selection.** The first step in our framework is to decide which layer of the LVLM should be used as the basis for attention weights. As shown in Fig. 4, the distribution of attention weights on image tokens varies across different layers and varies from model to model. For example, InstructBLIP (Dai et al., 2024) shows increasing attention levels in the later layers, whereas LLaVA-1.5 (Liu et al., 2023c) exhibits a concentration of attention in the earlier layers. To adapt these diverse models, we initially focus on selecting layers that exhibit a high proportion of image-related attention. Formally, we define the attention weight matrix for $i$-th layer as follows:

$$\mathbf{A}_i = \left[ \mathbf{a}_{h,q,k}^i \right]_{(h,q,k)=(1,1,1)}^{(h,q,k)=(H,N+M,N+M)}, \tag{3}$$

where $\mathbf{a}_{h,q,k}^i$ represents the attention weight assigned by head $h$, for query $q$, to key $l_k$ in layer $i$. The model handles two types of tokens: image tokens ($\mathcal{V} \in \mathbb{R}^{N \times D}$) and query tokens ($Q \in \mathbb{R}^{M \times D}$). Next, we calculate the proportion of attention dedicated to image tokens for each layer $i$ as:

$$AP_i^{\text{layer}} = \frac{\sum_h \sum_{k=1}^N \mathbf{a}_{h,(N+M),k}^i}{\sum_{i,h} \sum_{k=1}^N \mathbf{a}_{h,(N+M),k}^i}, \tag{4}$$

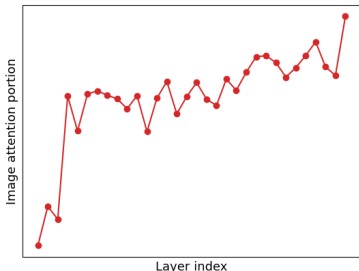

(a) InstructBLIP (Dai et al., 2024)

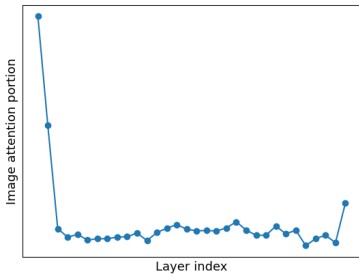

(b) LLaVA-1.5 (Liu et al., 2023c)

Figure 4: **Layer-wise image attention proportion in LVLMs.** This shows the proportion of attention given to image tokens at each layer relative to total attention. Different layers exhibit distinct attention patterns, which vary across models. Attention weights are averaged over 60 questions from the LLaVA-Bench (Liu et al., 2023c).

where $H$ is the total number of attention heads, $N$ is the number of image tokens, and $M$ is the number of query tokens. We sort the layers by this proportion and employ top-P sampling based on a predefined threshold value $\gamma$. The selected layers are:

$$\{\text{Selected Layers}\} = \text{top-P}(\{AP_i^{\text{layer}}\}_{i=1}^L, \gamma). \tag{5}$$

Here, top-P selects layers until the cumulative proportion of image attention across these layers meets or exceeds $\gamma$. These selected layers are used to analyze and adjust the attention at the token level and identify specific image tokens that the model may over-rely on, *i.e.*, *blind tokens*. While we leave the layer selection to be flexible by adjusting the $\gamma$ parameter, AvisC is not sensitive to $\gamma$ values (see Tab. 5), thus layers can be fixed in practice for simplicity.

**Blind token identification.** After selecting relevant layers, we calculate the attention weights for each image token within these layers. The attention proportion for image tokens, denoted as $AP^{\text{image}}$, is calculated by averaging the attention weights across the selected layers and attention heads:

$$AP^{\text{image}} = \frac{\sum_{i \in \{\text{Selected Layers}\}} \sum_{h=1}^H \mathbf{a}_{h,(N+M),[1:N]}^i}{|\{\text{Selected Layers}\}| \times H}. \tag{6}$$

To identify tokens that disproportionately capture the model's attention, *i.e.*, *blind tokens*, we calculate the mean ($\mu$) and standard deviation ($\sigma$) of the image attention weights. Tokens with an attention proportion exceeding $\mu + \lambda\sigma$ (where $\lambda$ is a hyperparameter) are classified as blind tokens:

$$\{\text{Blind Token Indices}\} = \{j | AP_j^{\text{image}} > \mu + \lambda\sigma\}. \tag{7}$$

**Contrastive decoding.** Our method seeks to reduce the influence of blind tokens, thereby decreasing the incidence of hallucinations in LVLMs. Drawing inspiration from recent successes in contrastive decoding (Leng et al., 2023; Favero et al., 2024), which effectively minimizes hallucinations by contrasting the differences between an image and its distorted counterpart, we adopt a similar scheme. We construct a new set of visual tokens $\mathcal{V}^*$ by zeroing out non-blind tokens and only leaving blind tokens, which biases the input towards emphasizing blind tokens:

$$\mathcal{V}^* = \bigcup_{j=1}^N \mathbb{1}_{\{j \in \text{Blind Token Indices}\}}(j)v_j. \tag{8}$$

Next, we compute the logits using both the original input ($\mathcal{V}$) and the biased input ($\mathcal{V}^*$):

$$\begin{aligned}
\ell_t &= \log p(\xi_t | \mathcal{V}, \mathcal{Q}, \xi_{<t}; \theta), \\
\ell_t^* &= \log p(\xi_t | \mathcal{V}^*, \mathcal{Q}, \xi_{<t}; \theta),
\end{aligned} \tag{9}$$

where $\ell_t$ and $\ell_t^*$ are the logits computed from the original and the biased inputs, respectively. We adjust the logits by contrasting the original and biased outputs, and then sample the next token $\xi_t$ from the following softmax distribution:

$$\xi_t \sim \text{Softmax}((1+\alpha)\ell_t - \alpha\ell_t^*). \tag{10}$$

Here, $\alpha$ is a hyperparameter that moderates the contrastive effect. This balances the distribution of attention across tokens thereby mitigating the likelihood of visual hallucinations in LVLMs.

## 4 EXPERIMENTS

### 4.1 EVALUATION SETUP

In our experiments, we did not constrain the LVLMs to provide one-word answers in discriminative tasks, which often require simple 'Yes' or 'No' responses. For instance, we avoid instructions such as "Please answer in one word." in the query text. We see that imposing a one-word response constraint on LVLMs leads to notable changes in performance (see Appendix D). For the experiments, we set P = 0.5 in Eq. (5), $\lambda = 1$ Eq. (7), $\alpha = 3$ for InsturctBLIP (Dai et al., 2024) and $\alpha = 2.5$ for LLaVA-1.5 (Liu et al., 2023c) in Eq. (10). [1] [2] [3]

---

[1]Visualization and analysis on blind tokens are shown in Appendix B.

[2]Further experimental and implementation details are in Appendix C.

[3]Additional experimental results can be found in Appendix D.

Table 1: **POPE benchmark results.** AVISC consistently outperforms *base* decoding and other methods: VCD (Leng et al., 2023) and M3ID (Favero et al., 2024). We reimplemented VCD and M3ID in our evaluation setup.

| Setup | | Method | InstructBLIP (Dai et al., 2024) | | | | LLaVA 1.5 (Liu et al., 2023c) | | | |
|---|---|---|---|---|---|---|---|---|---|---|
| | | | Acc. | Prec. | Rec. | F1 | Acc. | Prec. | Rec. | F1 |
| **MS-COCO (Lin et al., 2014)** | Random | *base* | 82.27 | 82.84 | 81.40 | 82.11 | 84.47 | 83.35 | 86.13 | 84.72 |
| | | VCD | 83.37 | 83.39 | 82.60 | 83.24 | 84.80 | 83.00 | 87.53 | 85.20 |
| | | M3ID | 84.37 | 84.62 | 84.00 | 84.31 | 86.00 | 85.11 | 87.27 | 86.18 |
| | | AVISC | 88.73 | 93.88 | 82.87 | 88.03 | 87.93 | 88.24 | 87.53 | 87.88 |
| | Popular | *base* | 77.77 | 74.81 | 83.73 | 79.02 | 82.23 | 79.72 | 86.47 | 82.95 |
| | | VCD | 78.00 | 75.12 | 83.73 | 79.19 | 82.27 | 79.19 | 87.53 | 83.15 |
| | | M3ID | 77.30 | 74.10 | 83.93 | 78.71 | 82.83 | 79.62 | 88.27 | 83.72 |
| | | AVISC | 83.90 | 81.33 | 88.00 | 84.53 | 84.33 | 81.71 | 88.47 | 84.96 |
| | Adversarial | *base* | 73.13 | 69.41 | 82.60 | 75.46 | 77.10 | 72.57 | 87.13 | 79.19 |
| | | VCD | 75.87 | 72.85 | 82.47 | 77.36 | 76.10 | 71.50 | 86.80 | 78.41 |
| | | M3ID | 76.03 | 72.47 | 83.93 | 77.79 | 77.70 | 73.23 | 87.33 | 79.66 |
| | | AVISC | 81.57 | 80.37 | 83.53 | 81.92 | 77.53 | 72.82 | 87.87 | 79.64 |
| **A-OKVQA (Schwenk et al., 2022)** | Random | *base* | 81.00 | 77.71 | 86.93 | 82.06 | 82.73 | 77.43 | 92.40 | 84.26 |
| | | VCD | 81.73 | 78.67 | 87.07 | 82.66 | 81.30 | 75.45 | 92.80 | 83.23 |
| | | M3ID | 82.33 | 77.81 | 90.47 | 83.66 | 83.57 | 77.86 | 93.80 | 85.09 |
| | | AVISC | 88.47 | 87.66 | 89.53 | 88.59 | 84.60 | 79.29 | 93.67 | 85.88 |
| | Popular | *base* | 75.00 | 70.14 | 87.07 | 77.69 | 76.10 | 69.86 | 91.80 | 79.34 |
| | | VCD | 75.33 | 70.52 | 87.07 | 77.92 | 75.43 | 68.58 | 93.87 | 79.26 |
| | | M3ID | 75.60 | 70.40 | 88.33 | 78.36 | 76.80 | 70.20 | 93.13 | 80.06 |
| | | AVISC | 81.77 | 77.82 | 88.87 | 82.98 | 78.83 | 72.10 | 94.07 | 81.63 |
| | Adversarial | *base* | 68.80 | 63.57 | 88.07 | 73.84 | 67.90 | 62.11 | 91.80 | 74.09 |
| | | VCD | 69.70 | 64.54 | 87.47 | 74.27 | 67.43 | 61.50 | 93.20 | 74.11 |
| | | M3ID | 69.57 | 64.21 | 88.40 | 74.39 | 68.10 | 61.99 | 93.60 | 74.58 |
| | | AVISC | 72.53 | 67.12 | 88.33 | 76.28 | 68.97 | 62.70 | 93.67 | 75.11 |
| **GQA (Hudson & Manning, 2019)** | Random | *base* | 80.00 | 77.08 | 85.40 | 81.02 | 82.40 | 77.03 | 92.33 | 83.99 |
| | | VCD | 81.73 | 79.35 | 85.80 | 82.45 | 82.27 | 75.85 | 94.67 | 84.22 |
| | | M3ID | 80.57 | 76.77 | 87.67 | 81.85 | 82.83 | 76.64 | 94.47 | 84.62 |
| | | AVISC | 86.47 | 85.89 | 87.27 | 86.57 | 85.00 | 78.81 | 95.73 | 86.45 |
| | Popular | *base* | 73.53 | 68.80 | 86.13 | 76.49 | 72.03 | 65.57 | 92.80 | 76.84 |
| | | VCD | 74.10 | 69.45 | 86.07 | 76.87 | 71.77 | 64.90 | 94.80 | 77.05 |
| | | M3ID | 74.57 | 69.45 | 87.83 | 77.53 | 72.83 | 66.04 | 94.00 | 77.58 |
| | | AVISC | 78.00 | 73.68 | 87.13 | 79.84 | 74.80 | 67.46 | 95.80 | 79.17 |
| | Adversarial | *base* | 68.00 | 63.49 | 84.73 | 72.59 | 68.73 | 62.54 | 93.40 | 74.92 |
| | | VCD | 70.27 | 65.43 | 85.93 | 74.29 | 68.27 | 62.00 | 94.40 | 74.84 |
| | | M3ID | 68.90 | 64.06 | 86.13 | 73.47 | 68.13 | 61.88 | 94.47 | 74.78 |
| | | AVISC | 73.07 | 67.80 | 87.87 | 76.54 | 69.20 | 62.61 | 95.33 | 75.58 |

**LVLMs.** We evaluated AVISC on two state-of-the-art LVLMs: **LLaVA-1.5** (Liu et al., 2023c) and **InstructBLIP** (Dai et al., 2024), both incorporating Vicuna 7B (Chiang et al., 2023) as an LLM backbone. LLaVA-1.5 synchronizes image and text modalities by applying linear projection layers, while InstructBLIP uses the Q-Former (Li et al., 2023a) to efficiently link visual and textual features using a fixed number of tokens (*e.g.*, 32 tokens). Notably, AVISC is model-agnostic and can integrate with various of LVLM architectures.

**Benchmarks. (1) POPE** (Li et al., 2023c) views hallucination evaluation as a binary classification task (yes/no) with questions regarding object presence (*e.g.*, "Is there a cat in the image?"). It includes 500 images from MS-COCO and evaluates them based on visible objects and imaginary ones across different object categories, using three setups (random, popular, and adversarial). **(2) MME** (Fu et al., 2024) evaluates 14 subtasks including object hallucination by answering binary questions about object existence, count, position, color, *etc*. **(3) AMBER** (Wang et al., 2023b) includes both generative and discriminative tasks, focusing on hallucinations related to object existence, attributes, and relationships, with performance evaluated using CHAIR for generative tasks and an F1 score for discriminative tasks. The overall AMBER score is calculated as $((100 - \text{CHAIR}) + \text{F1})/2$.

**Baselines.** AVISC aims to minimize hallucinations in LVLMs without the need for external models, costly self-feedback mechanisms, or further training. We select baseline methods that fulfill these

Table 2: **MME-Hallucination (Fu et al., 2024) benchmark results.** Our method effectively reduces hallucinations at both object and attribute levels, surpassing VCD (Leng et al., 2023) and M3ID (Favero et al., 2024) in Total Score.

| Model | Method | Object-level | | Attribute-level | | Total Score |
|---|---|---|---|---|---|---|
| | | Existence | Count | Position | Color | |
| **InstructBLIP** | *base* | $170.19_{(\pm11.12)}$ | $89.52_{(\pm11.04)}$ | $67.62_{(\pm14.04)}$ | $114.76_{(\pm9.60)}$ | $442.09_{(\pm31.51)}$ |
| | VCD | $172.62_{(\pm8.92)}$ | $98.33_{(\pm15.99)}$ | $71.90_{(\pm13.42)}$ | $117.14_{(\pm10.70)}$ | $459.99_{(\pm16.56)}$ |
| | M3ID | $173.89_{(\pm10.52)}$ | $89.72_{(\pm13.44)}$ | $72.72_{(\pm14.77)}$ | $110.56_{(\pm7.20)}$ | $446.88_{(\pm28.54)}$ |
| | **AVISC** | $184.76_{(\pm5.56)}$ | $82.85_{(\pm12.16)}$ | $74.76_{(\pm6.19)}$ | $131.43_{(\pm4.76)}$ | $473.80_{(\pm19.67)}$ |
| **LLaVA 1.5** | *base* | $173.57_{(\pm8.16)}$ | $110.00_{(\pm15.82)}$ | $100.47_{(\pm18.78)}$ | $125.24_{(\pm15.91)}$ | $509.28_{(\pm30.57)}$ |
| | VCD | $172.14_{(\pm8.09)}$ | $117.14_{(\pm8.76)}$ | $103.33_{(\pm20.56)}$ | $119.52_{(\pm8.58)}$ | $512.14_{(\pm31.82)}$ |
| | M3ID | $178.33_{(\pm6.83)}$ | $107.22_{(\pm14.78)}$ | $96.39_{(\pm5.52)}$ | $127.50_{(\pm8.28)}$ | $509.44_{(\pm22.52)}$ |
| | **AVISC** | $189.29_{(\pm1.82)}$ | $104.76_{(\pm11.66)}$ | $106.19_{(\pm13.93)}$ | $127.86_{(\pm9.13)}$ | $528.09_{(\pm24.70)}$ |

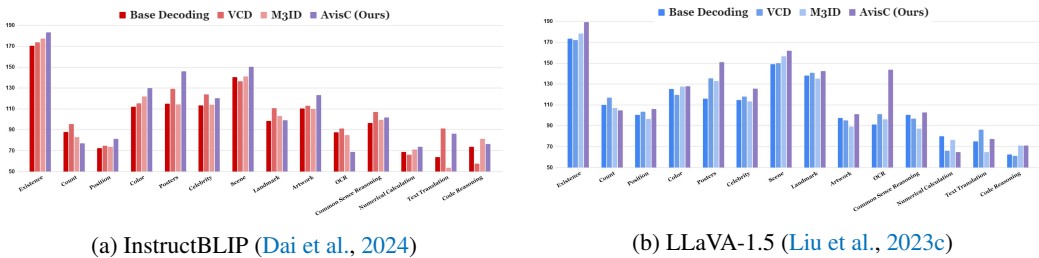

(a) InstructBLIP (Dai et al., 2024)        (b) LLaVA-1.5 (Liu et al., 2023c)

Figure 5: **Performance comparison on MME-Fullset.** AVISC achieves top performance in 7 of 14 categories with InstructBLIP (Dai et al., 2024) and in 11 categories with LLaVA-1.5 (Liu et al., 2023c). Beyond minimizing hallucinations, AVISC also boosts the general functionality of LVLMs.

conditions. We choose recent contrastive decoding methods as baselines, notably **VCD** (Leng et al., 2023) and **M3ID** (Favero et al., 2024). These methods are designed to reduce object hallucinations by enhancing the influence of the reference image over the language model's prior or statistical bias, by contrasting output distributions from both original and altered visual inputs. We reimplemented VCD and M3ID within our evaluation framework.

## 4.2 RESULTS ON BENCHMARKS

**POPE.** Table 1 showcases the performance of different methods on the POPE benchmark (Li et al., 2023c) across MS-COCO (Lin et al., 2014), A-OKVQA (Schwenk et al., 2022), and GQA (Hudson & Manning, 2019) datasets, evaluated under Random, Popular, and Adversarial setups. (**AVISC**) consistently outperforms the baseline (*base*) and other decoding methods (VCD (Leng et al., 2023), M3ID (Favero et al., 2024)) in most cases, achieving the highest Accuracy and F1 scores. It also demonstrates balanced improvements in Precision and Recall, indicating a reduction in errors and better information capture. For InstructBLIP, AVISC shows a significant performance boost, particularly in mitigating hallucinations related to object existence. However, LLaVA 1.5 exhibits less pronounced improvements in Popular and Adversarial setups, highlighting its limitations in more challenging scenarios. Yet, overall, AVISC proves to be robust and effective across different datasets and query setups.

**MME-Hallucination.** Table 2 presents performance results for InstructBLIP (Dai et al., 2024) and LLaVA 1.5 (Liu et al., 2023c) on the MME-Hallucination benchmark (Fu et al., 2024), focusing on object-level (Existence, Count) and attribute-level (Position, Color) metrics. Both models exhibit significant improvements in the Existence category with Ours, achieving the highest scores. While VCD (Leng et al., 2023) performs best in the Count metric, AVISC excels in the Position and Color categories, attaining the top scores for both models. AVISC demonstrates superior performance in Total Score compared to other methods, affirming its effectiveness in reducing hallucinations and improving accuracy across multiple metrics.

**MME-Fullset.** Figure 5 compares the performance of various decoding methods on the MME-Fullset (Fu et al., 2024) across 14 categories. AVISC generally outperforms other methods, achieving

Table 3: **AMBER (Wang et al., 2023b) benchmark results.** AVISC outperforms contrastive decoding baselines (Leng et al., 2023; Favero et al., 2024) in both generative and discriminative tasks, achieving the highest AMBER score.

| Metric | | InstructBLIP (Dai et al., 2024) | | | | LLaVA 1.5 (Liu et al., 2023c) | | | |
|---|---|---|---|---|---|---|---|---|---|
| | | *base* | VCD | M3ID | **AVISC** | *base* | VCD | M3ID | **AVISC** |
| **Generative** | CHAIR↓ | $8.40_{(\pm0.57)}$ | $7.60_{(\pm0.42)}$ | $6.85_{(\pm0.07)}$ | $6.70_{(\pm0.28)}$ | $7.95_{(\pm0.64)}$ | $6.70_{(\pm0.42)}$ | $6.00_{(\pm0.14)}$ | $6.25_{(\pm0.07)}$ |
| | Cover↑ | $46.40_{(\pm1.27)}$ | $47.65_{(\pm0.35)}$ | $47.20_{(\pm0.71)}$ | $46.65_{(\pm1.48)}$ | $44.45_{(\pm0.21)}$ | $46.50_{(\pm0.28)}$ | $48.90_{(\pm0.28)}$ | $46.55_{(\pm0.64)}$ |
| | Hal↓ | $31.10_{(\pm0.64)}$ | $29.90_{(\pm0.99)}$ | $27.50_{(\pm0.71)}$ | $28.00_{(\pm0.28)}$ | $31.00_{(\pm2.83)}$ | $27.80_{(\pm1.70)}$ | $26.00_{(\pm0.28)}$ | $25.60_{(\pm1.70)}$ |
| | Cog↓ | $2.60_{(\pm0.05)}$ | $2.20_{(\pm0.14)}$ | $2.20_{(\pm0.14)}$ | $2.55_{(\pm0.35)}$ | $2.15_{(\pm0.35)}$ | $1.95_{(\pm0.35)}$ | $1.45_{(\pm0.07)}$ | $2.00_{(\pm0.04)}$ |
| **Discriminative** | Acc.↑ | $68.20_{(\pm0.14)}$ | $69.65_{(\pm0.35)}$ | $69.05_{(\pm0.35)}$ | $72.60_{(\pm0.42)}$ | $67.00_{(\pm0.71)}$ | $67.30_{(\pm1.41)}$ | $67.25_{(\pm0.21)}$ | $70.70_{(\pm0.57)}$ |
| | Prec.↑ | $79.00_{(\pm0.14)}$ | $80.70_{(\pm0.42)}$ | $79.70_{(\pm0.28)}$ | $72.60_{(\pm0.42)}$ | $85.45_{(\pm0.49)}$ | $86.10_{(\pm1.70)}$ | $86.50_{(\pm0.57)}$ | $85.45_{(\pm0.21)}$ |
| | Rec.↑ | $70.70_{(\pm0.42)}$ | $71.60_{(\pm0.42)}$ | $71.25_{(\pm0.35)}$ | $76.10_{(\pm0.05)}$ | $60.95_{(\pm1.20)}$ | $60.55_{(\pm1.34)}$ | $60.05_{(\pm0.07)}$ | $67.55_{(\pm0.92)}$ |
| | F1↑ | $74.60_{(\pm0.14)}$ | $75.90_{(\pm0.42)}$ | $75.25_{(\pm0.07)}$ | $78.60_{(\pm0.28)}$ | $71.10_{(\pm0.99)}$ | $71.10_{(\pm1.56)}$ | $70.90_{(\pm0.14)}$ | $75.45_{(\pm0.64)}$ |
| **AMBER↑** | | $83.10_{(\pm0.35)}$ | $84.15_{(\pm0.05)}$ | $84.20_{(\pm0.07)}$ | $85.95_{(\pm0.05)}$ | $81.58_{(\pm0.18)}$ | $82.20_{(\pm0.99)}$ | $82.45_{(\pm0.14)}$ | $84.60_{(\pm0.35)}$ |

(a) InstructBLIP (Dai et al., 2024)  (b) LLaVA-1.5 (Liu et al., 2023c)

Figure 6: **Performance comparison on AMBER discriminative tasks.** Our demonstrates superior performance overall, particularly excelling in the Existence and Action categories in both Instruct-BLIP (Dai et al., 2024) and LLaVA-1.5 (Liu et al., 2023c). Detailed results are in Appendix D.6.

top performance in 7 categories for InstructBLIP and 11 categories for LLaVA 1.5. This demonstrates AVISC's effectiveness in enhancing understanding of visual information through attention calibration. However, both models see a decline in performance for the Count category with AVISC, and InstructBLIP shows poor OCR performance. Conversely, LLaVA 1.5 experiences significant OCR improvement with AVISC, indicating the method's variable impact across different models. Overall, AVISC provides consistent and superior results across most tasks compared to other methods.

**AMBER.** Table 3 presents the results of InstructBLIP (Dai et al., 2024) and LLaVA 1.5 (Liu et al., 2023c) on the AMBER benchmark (Wang et al., 2023b), which includes both generative tasks (detailed image descriptions) and discriminative tasks (answering questions about images). Both models show significant improvements in Accuracy and F1 scores in discriminative tasks using AVISC, outperforming the base, VCD (Leng et al., 2023), and M3ID (Favero et al., 2024) methods. In generative tasks, AVISC continues to exhibit substantial gains, indicating its effectiveness in generating detailed image descriptions. Notably, there is a marked improvement in the Existence metric, highlighting the method's accuracy in detecting objects. Overall, both models achieve the highest performance across most metrics with AVISC. AVISC stands out with the highest AMBER score, indicating its comprehensive superiority in both generative and discriminative tasks. Fig. 6 visualizes the performance of each decoding method across discriminative tasks in the AMBER benchmark.

## 4.3 ABLATION STUDY

**Ablations on $\alpha$ and $\lambda$.** $\lambda$ is a threshold for identifying blind tokens that excessively concentrate attention weights, as detailed in Eq. (7). On the other hand, $\alpha$ is a contrastive decoding hyperparameter, defined in Eq. (10). We conduct ablation experiments on the MME-Hallucination (Liu et al., 2023d)

Table 4: $\alpha$ and $\lambda$ **ablations on MME-Hallucination** (Fu et al., 2024). We set $\alpha = 3$, $\lambda = 1$ for InstructBLIP (Dai et al., 2024) and $\alpha = 2.5$, $\lambda = 1$ for LLaVA-1.5 (Liu et al., 2023b).

| (a) InstructBLIP (Dai et al., 2024) ($\lambda = 1$) | | | | | | (b) InstructBLIP (Dai et al., 2024) ($\alpha = 3$) | | | | | | (c) LLaVA-1.5 (Liu et al., 2023b) ($\lambda = 1$) | | | | | |
|---|---|---|---|---|---|---|---|---|---|---|---|---|---|---|---|---|---|
| | Object | | Attribute | | Total Score | | Object | | Attribute | | Total Score | | Object | | Attribute | | Total Score |
| $\alpha$ | Exist. | Count | Position | Color | | $\lambda$ | Exist. | Count | Position | Color | | $\alpha$ | Exist. | Count | Position | Color | |
| 0.5 | 180 | 83.33 | 80.00 | 130 | 473.33 | 0.0 | 180 | 75.00 | 60.00 | 115.00 | 430.00 | 0.5 | 185 | 111.66 | 103.33 | 115.00 | 514.99 |
| 2.0 | 180 | 86.66 | 75.00 | 135 | 476.66 | 0.1 | 185 | 60.00 | 65.00 | 123.33 | 433.33 | 2.0 | 180 | 103.33 | 101.66 | 120.00 | 504.99 |
| 2.5 | 180 | 85.00 | 71.66 | 135 | 471.66 | **1.0** | **195** | **75.00** | **73.33** | **135.00** | **478.33** | **2.5** | **180** | **105.00** | **111.66** | **120.00** | **516.66** |
| **3.0** | **195** | **75.00** | **73.33** | **135** | **478.33** | 1.5 | 195 | 75.00 | 73.33 | 135.00 | 478.33 | 3.0 | 180 | 105.00 | 111.66 | 120.00 | 516.66 |

Table 5: $\gamma$ **ablations** on **(a)** POPE-COCO-Random and **(b)** MME-Hallucination benchmarks.

| (a) LLaVA-1.5 ($\lambda = 1$, $\alpha = 2.5$) | | | | | (b) LLaVA-1.5 ($\lambda = 1$, $\alpha = 2.5$) | | | | | |
|---|---|---|---|---|---|---|---|---|---|---|
| $\gamma$ | Acc. | Prec. | Rec. | F1 | $\gamma$ | Existence | Count | Position | Color | Total Score |
| **0.5 (Ours)** | 87.93 | 88.24 | 87.53 | 87.88 | **0.5 (Ours)** | 189.29 | 104.76 | 106.19 | 127.86 | 528.10 |
| 0.1 | 86.77 | 83.98 | 90.87 | 87.29 | 0.1 | 167.50 | 101.80 | 103.33 | 117.50 | 490.13 |
| 0.3 | 87.47 | 85.35 | 90.47 | 87.83 | 0.3 | 180.00 | 98.33 | 114.16 | 125.00 | 517.49 |
| 1.0 | 88.27 | 88.06 | 88.53 | 88.30 | 1.0 | 182.50 | 108.33 | 109.99 | 117.50 | 518.32 |

benchmark to evaluate how these hyperparameters influence the performance of our AVISC. Tab. 4 (a) and (c) show the experimental results using InstructBLIP (Dai et al., 2024) and LLaVA-1.5 (Liu et al., 2023c), respectively, where we fixed $\lambda$=1 and varied $\alpha$ from 0.5 to 3. While there are variations across evaluation categories, overall performance consistently improves with increasing values of $\alpha$. Specifically, each LVLM achieves the highest total score at $\alpha$=3 and $\alpha$=2.5. These results suggest that enhancing the intensity of contrastive decoding can improve the robustness of LVLMs against visual hallucinations. Tab. 4 (b) presents the experimental results for the InstructBLIP (Dai et al., 2024) model using varying values of $\lambda$. The results indicate that performance enhances as $\lambda$ increases, demonstrating that our AVISC yields better results when applied to a smaller number of blind tokens with excessive attention weight.

**Ablations on $\gamma$.** We conduct ablative experiments with the LLaVA-v1.5-7b model to explore the sensitivity to the $\gamma$ parameter. We explore the sensitivity of the model to the $\gamma$ parameter while fixing $\lambda = 1.0$ and $\alpha = 2.5$. The results, shown in Tab. 5, indicate that performance remains robust across a range of $\gamma$ values, except for extreme settings like $\gamma = 0.1$. In (a), $\gamma = 0.5$, our default, achieves high accuracy and balanced metrics on POPE-COCO-Random, while in (b), it delivers the highest total score in the MME-Hallucination benchmark. Overall, our experiments demonstrate that the impact of these parameters on performance is minimal, thus reducing the need for extensive tuning. Therefore, we fixed $\lambda = 1.0$ and $\gamma = 0.5$ during our experiments.

# 5 CONCLUSION

This study highlights the phenomenon of *blind tokens* in LVLMs, where the attention mechanism disproportionately focuses on uninformative image tokens, leading to skewed attention distributions and contributing to hallucinatory outputs. To mitigate this issue, we introduced a novel decoding technique, termed Attentional Vision Calibration (AVISC), which dynamically adjusts the logits by identifying blind tokens through an analysis of image-wise attention distribution. AVISC recalibrates the model's attention, without requiring additional retraining, external datasets, or self-feedback mechanisms, thereby significantly reducing the model's reliance on these blind tokens. Through extensive experiments on hallucination benchmarks such as POPE, MME, and AMBER, AVISC consistently outperforms existing decoding techniques, improving the model's ability to capture subtle visual details by redirecting attention toward more informative tokens.

**Limitation.** The discriminative capabilities of LVLMs using AVISC diminish in tasks that involve counting objects within an image. This suggests that blind tokens may sometimes contain essential information, particularly in object-counting scenarios, leading to reduced performance in the "Count" category of MME and the "Numbers" category of AMBER.

**Future Work.** Building upon insights from (Darcet et al., 2023), the blind token phenomenon may extend beyond LVLMs and could be a general characteristic of transformer-based architectures. This motivates us to reasonably hypothesize that such counterintuitive attentional biases might be inherent to attention mechanisms themselves. We aim to further explore this in future research.

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

# APPENDIX

## CONTENTS

## A    COMPARISON OF OUR WORK WITH (DARCET ET AL., 2023)

**Short summary of (Darcet et al., 2023).**

Darcet et al. (2023) identify artifacts in feature maps of various vision transformers, particularly when comparing DINOv1 (Caron et al., 2021) and DINOv2 (Oquab et al., 2023). Darcet et al. observe that the image token attention weights are not evenly distributed across informative regions of the image; instead, they are concentrated in seemingly unnecessary regions, such as redundant background areas. These regions correspond to artifact patches, referred to as "high-norm outlier tokens", which receive high attention.

Darcet et al. demonstrate that high-norm outlier tokens contain minimal local information while retaining significant global information. Specifically, in position prediction and pixel reconstruction tasks – where locality is crucial – these outlier tokens perform significantly worse than normal image tokens. However, when Darcet et al. conduct linear probing classification experiments using the class token, high-norm outlier tokens, and normal image tokens, the high-norm outlier tokens outperform the normal image tokens. This indicates that while these outlier tokens lose local information, they effectively encode global information about the image. Darcet et al. suggests that during the attention mechanism's internal operations, these tokens are repurposed to capture global information, likely due to their association with repetitive or less informative image patches. Moreover, the paper presents experimental results demonstrating that when additional memory space (*i.e.*, register tokens) is added to store this information, the artifacts disappear.

**How different it is from Blind tokens?**

"Blind tokens" in our work and the "high-norm outlier tokens" described in (Darcet et al., 2023) show similar findings. Both tokens are identified by their significantly high attention weights and are associated with regions in the image that appear irrelevant to the target task. However, despite these conceptual similarities, there are several key differences between the two:

- Source of Attention Weights: In (Darcet et al., 2023), the high-norm outlier tokens are derived from the attention weights computed within the vision transformer layers, whereas our blind tokens are based on attention weights calculated by the LLMs within the Large Vision-Language Models (LVLMs) (*e.g.* Vicuna-7B in LLaVA-v1.5-7B). Furthermore, each transformer-based architecture has different mask designs.

- Task & Architectural Differences: The presence and pattern of high-norm tokens in vision transformers appear to be highly sensitive to the training schemes and specific architectures employed. For example, while DINOv1 (Caron et al., 2021) does not exhibit high-norm tokens, they suddenly emerge in DINOv2 (Oquab et al., 2023), which has been refined for dense prediction tasks. This suggests that the patterns of high-norm tokens can vary significantly depending on the trained target task and the model architecture. Given that these models are all confined to visual tasks, this variation is likely influenced by the specific task the model is trained on. LVLMs are essentially a combination of LLM, a visual encoder, and a vision-to-text projector, primarily designed for image-related question-answering tasks. These models are trained using an auto-regressive token prediction scheme, which significantly differs from the architecture and training methods of vision transformers. Additionally, Darcet et al. shows that high-norm tokens in vision transformers encapsulate global image information, as evidenced by their strong performance in linear probing classification tasks. However, it remains unclear whether these tokens play a similarly critical role in the language token prediction tasks of LVLMs.

- Domain Differences: While both our blind tokens and the high-norm tokens (Darcet et al., 2023) originate from image patch tokens, LVLMs project image tokens into a LLM space for further attention computations. This contrasts with vision transformers, where all tokens remain within the image patch domain during attention operations. It is uncertain whether the high-norm tokens defined in (Darcet et al., 2023) can be similarly defined in LVLMs. To our knowledge, we first observe and define a phenomenon akin to this in LVLMs.

To explore the correlation between these concepts further, we conducted additional experiments. Specifically, when examining the vision transformer used as the visual encoder in LLaVA-1.5-7B (*e.g.*, CLIP-L-336px), we observed the emergence of high attention weight regions, corresponding

to the high-norm tokens described in (Darcet et al., 2023). Furthermore, we discovered a notable correlation between these high-norm tokens and the blind tokens selected based on LLM attention weights. According to our analysis on the POPE-COCO-Random benchmark, we found the following statistics (The criteria for determining high-norm tokens were the same as for blind tokens):

- $P(\text{blind token}|\text{high-norm token}) = 40.38\%$
- $P(\text{high-norm token}|\text{blind token}) = 31.27\%$

While high-norm tokens do not completely overlap with blind tokens, the relatively small number of blind and high-norm tokens (on average, 12.95 out of 576 total tokens) suggests a strong correlation. Therefore, these tokens may share certain underlying properties. However, many blind tokens are distributed at the beginning and end of the image token sequence. Whether high-norm tokens share this characteristic is unclear, but this feature appears to be unique to blind tokens in LVLMs, rather than high-norm tokens in the ViT-based architecture discussed in (Darcet et al., 2023). Based on this evidence, we conclude that while blind tokens and the artifacts described in (Darcet et al., 2023) are not identical, they may share certain properties. Despite these differences, we believe that (Darcet et al., 2023) complements our findings and could further stimulate exploration of erroneous focus within attention mechanisms across modern architectures.

**Is it really a good idea to reduce the dependency on blind tokens?**

In our work, we observed that, similar to (Darcet et al., 2023), blind tokens tend to emerge in non-important patches, such as backgrounds. However, a key difference is that while they describe high-norm tokens as containing global information at the expense of local information, we found that blind tokens contain information irrelevant to generating a response to the textual query. While Darcet et al. found that high-norm tokens carry global information, they try to mitigate their influence, as their presence negatively impacts dense prediction tasks that require spatial locality. For instance, in attention-based object discovery tasks like LOST (Siméoni et al., 2021), high-norm tokens can directly cause errors. Similarly, we argue that the presence of blind tokens is not a desirable phenomenon, especially for image-related response prediction tasks. Our approach stems from the observation that blind tokens do not contain question-relevant information. If a small number of blind tokens carry global information and are key sources for token prediction (with these tokens, as expected, having high attention weights), their performance should be reasonably strong. However, the results shown in Figs. 1 and 2 challenge this assumption. As seen in Fig. 2, the truly important information is often found in non-blind tokens. Therefore, our contrastive decoding scheme, which reduces the influence of blind tokens and strengthens the influence of non-blind tokens, can be understood as a method to mitigate hallucination.

## B    VISUALIZATIONS & ANALYSIS ON BLIND TOKENS

### B.1    MORE EXAMPLES OF ATTENTION BIAS IN LLAVA

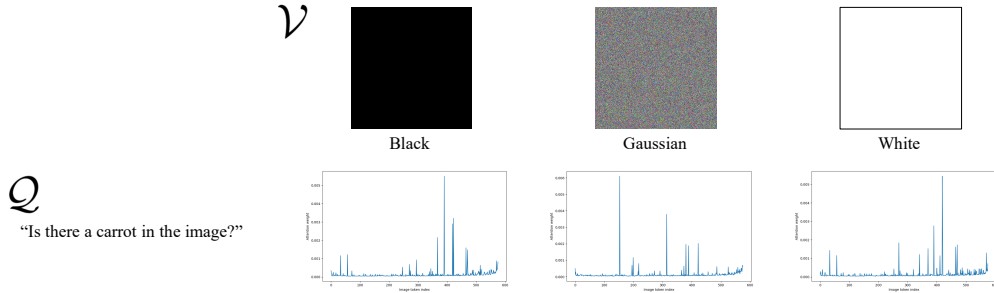

Figure 7: Attention weights for images without semantics nor query-related information.

Our method aims to correct the identified tendency (somewhat misaligned image attention patterns) in LVLMs without undermining the fundamental design of the attention mechanism itself. Blind tokens are image tokens that excessively occupy attention weights even though they do not significantly impact the prediction logits. Based on our observation (see Figs. 1 and 7) that the blind tokens tend

918
919
920
921
922
923
924
925
926

to miss the important semantics, and in fact, tokens with lower attention weights can contain such information, our approach aims to recalibrate the attention weights on the image. We do this by contrasting two probability distributions: one with all image tokens and one with only the blind tokens. In fact, our approach does not modify the intrinsic attention values themselves during this process. Instead, by contrasting the probability distributions, we reduce the effect of blind tokens and give more consideration to the rest of the tokens at the final token prediction phase. Through experiments, we verify our hypothesis on blind tokens and show that our method ensures a broader and more accurate semantic capture of the image, thereby reducing hallucinations.

### B.2 VISUALIZATION OF BLIND TOKENS AND TARGET OBJECTS

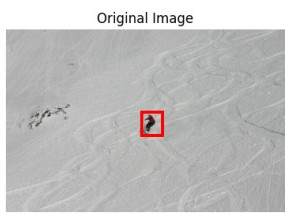 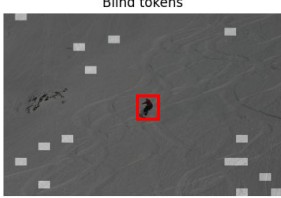

Q: "Is there a snowboard in the image?"

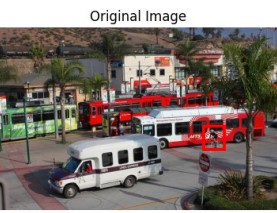 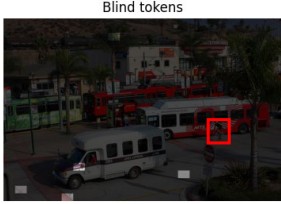

Q: "Is there a person in the image?"

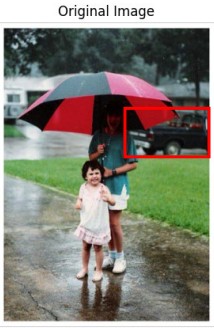 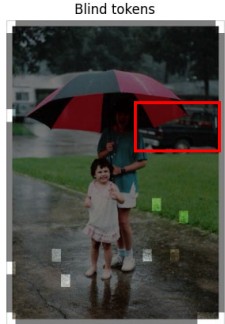

Q: "Is there a truck in the image?"

Figure 8: Visualization of blind tokens and target object bounding box on the POPE benchmark.

To support the claim that there is a mismatch between blind tokens and query-relevant information, we visualize blind tokens and the bounding boxes of the target objects in several images from the POPE-COCO benchmark in Fig. 8.

### B.3 DISTRIBUTIONS OF BOUNDING BOXES AND BLIND TOKENS

Additionally, we analyze the distribution of object bounding boxes and blind tokens in the POPE-COCO-Random benchmark, visualizing the results as heatmaps in Fig. 9. This shows that while object bounding boxes are evenly distributed around the center of the image, blind tokens tend to be concentrated at the edges of the image, which indicates a significant disparity in distribution.

### B.4 VISUALIZATION AND STATISTICS OF BLIND TOKENS

We conduct a correlation analysis between actual object region and attention weights in LVLMs using the 3000 COCO2014 validation dataset. The results are in Fig. 10. We asked LVLMs to describe

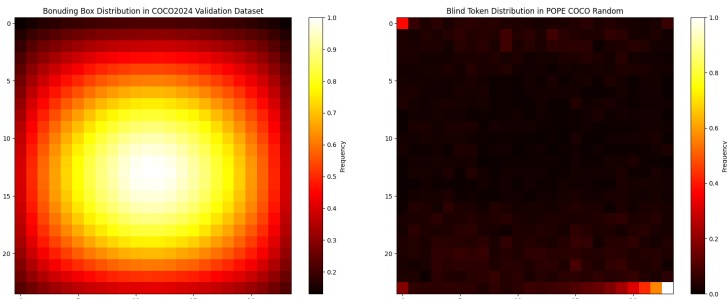

Figure 9: Average distributions of bounding boxes and blind tokens in COCO dataset.

the image and analyzed the attention distribution on $24 \times 24$ patches, comparing bounding boxes *vs.* attention weights on these patches. Particularly, we assess the proportion of blink tokens within the bounding boxes. The results show that, on average, only 3.7% of blind tokens overlapped with the actual object regions, and only 23.3% of the attention weights were assigned to the object regions while the rest were assigned to other regions. This indicates that although blind tokens receive excessive attention weights, they fail to effectively capture the objects that are crucial for accurate image description.

### B.5 HISTOGRAM OF BLIND TOKENS

Fig. 11 illustrates the histogram of the number of blind tokens identified by AVISC and the image token attention weights for these blind tokens when evaluating the LLaVA-1.5v-7B model on the POPE-COCO-Random benchmark. In this experimental setup, an average of 12.95 blind tokens appeared, accounting for 33.23% of the image token attention weight.

### B.6 BLIND TOKENS AND TOKEN PROBABILITY DISTRIBUTION

Fig. 12 visualizes the location of blind tokens for a given image and query, and presents the token logit values of both the baseline model and AVISC. For example, in the first problem, which asked whether there is a banana in the image, the original probability distribution was: 'No' at 89.62%, 'Yes' at 8.46%, and 'There' at 1.56%. After applying AVISC, the logit distribution shifted to: 'No' at 98.00%, 'There' at 1.35%, and 'Yes' at 0.61%.

## C MORE EXPERIMENTAL DETAILS

### C.1 FURTHER IMPLEMENTATION DETAILS

The text generation decoding process utilized cut-off sampling to assess the effectiveness of logit distribution enhancements achieved through AVISC. Following the experimental settings of VCD (Leng et al., 2023), tokens with probability values below $\beta$ times the maximum generating token probability were masked and excluded from sampling. Specifically, we only consider text tokens that belong to $\mathcal{H}$ at the generation step $t$:

$$\mathcal{H}(\xi_{<t}) = \{\xi_t \in \mathcal{H} : p\left(\xi_t \mid \mathcal{V}, \mathcal{Q}, \xi_{<t}; \theta\right) \geq \beta \max_{w} p\left(w \mid \mathcal{V}, \mathcal{Q}, \xi_{<t}; \theta\right)\}. \tag{11}$$

We set the balancing parameter $\beta$ to 0.1. We configured the LVLMs to generate a maximum of 64 tokens for both generative and discriminative tasks. During our experiments with the LLaVA-1.5 (Liu et al., 2023c), we utilized the "llava_v1" template provided by LLaVA for the conversation setup.

For reproducing the VCD (Leng et al., 2023), we referenced the official code provided by VCD. We set $\alpha$ for contrastive decoding to 1.0, the cut-off hyperparameter $\beta$ to 0.1, and the diffusion noise step $T$ used for generating noise images to 500. In the reproduction of the M3ID (Favero et al., 2024), we used 0.2 as the $\lambda$. The aforementioned token generation decoding method was utilized to ensure a fair comparison with other methods.

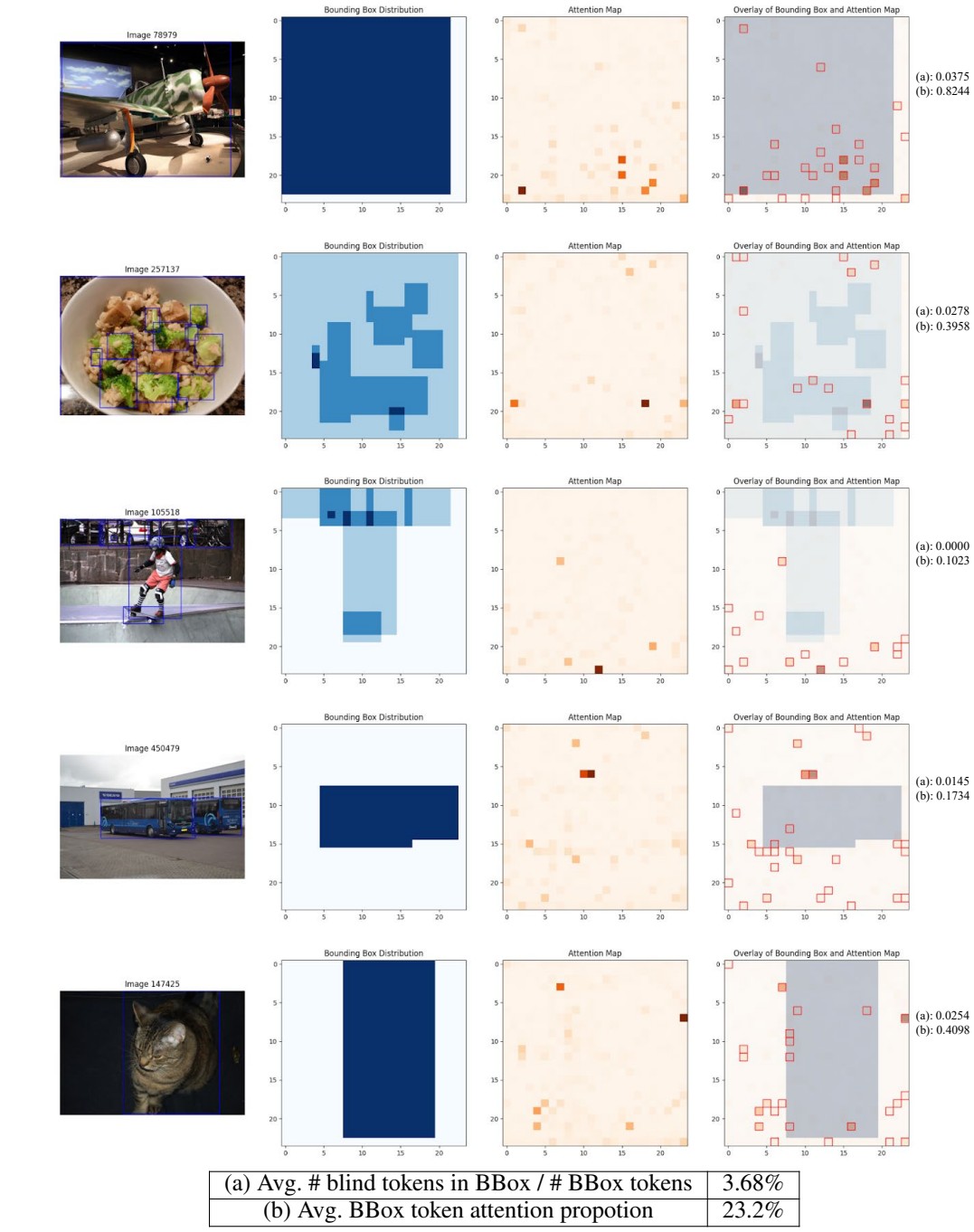

| (a) Avg. # blind tokens in BBox / # BBox tokens | 3.68% |
|---|---|
| (b) Avg. BBox token attention propotion | 23.2% |

Figure 10: Visualization and statistics of object bounding boxes and blind tokens on the COCO2014 dataset.

## C.2 EVALUATION BENCHMARKS

**POPE.** We employed the official benchmark described in (Li et al., 2023c), which comprises 3,000 question-answer pairs across the random, popular, and adversarial settings. Our queries followed the structure 'Is there a [object] in the image?', where [object] is selected either at random, from the most common objects in the dataset, or from objects that are often found alongside the specified [object], tailored to the random, popular, and adversarial scenarios, respectively. The model's effectiveness

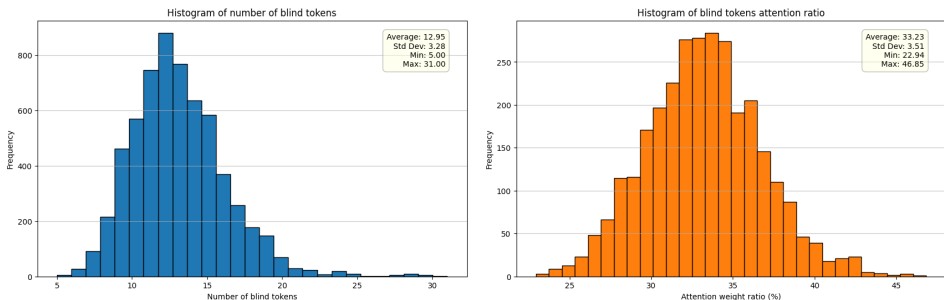

Figure 11: Number and attention weight statistics of blind tokens on POPE-COCO-Random benchmark.

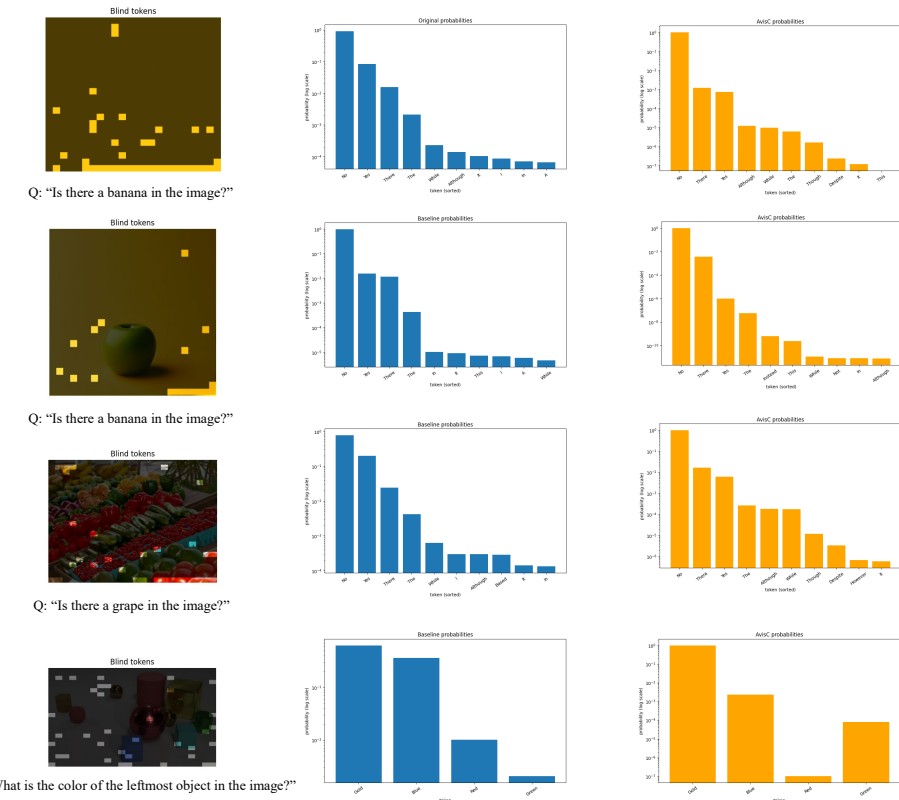

Figure 12: Visualization of blind tokens and logit probability enhancement by AvisC.

was assessed by determining if the model-generated response accurately matched the correct answer ('Yes' or 'No'), using metrics such as accuracy, precision, recall, and mean F1-score.[4]

**MME.**    The MME dataset (Fu et al., 2024) is divided into 10 perceptual categories (existence, count, position, color, posters, celebrity, scene, landmark, artwork, OCR) and four cognitive categories (commonsense reasoning, numerical calculation, text translation, code reasoning). While we utilized the official dataset, we modified the prompt by eliminating the instruction (*i.e.* "Answer the question using a single word or phrase.") that restricts LVLMs to response length.[5]

**AMBER.**    The AMBER dataset (Wang et al., 2023b) comprises 1004 images along with their associated generative task prompts (*i.e.* "Describe this image.") and questions categorized into three discriminative task types (existence, attribute, and relation). We randomly sampled 500 questions

---

[4]https://github.com/RUCAIBox/POPE
[5]https://github.com/BradyFU/Awesome-Multimodal-Large-Language-Models/tree/Evaluation

for the generative tasks and 5000 questions for the discriminative tasks, and the evaluation was established on official protocols.[6]

**LLaVA-Bench.** (Liu et al., 2023c) features a collection of 24 images, accompanying 60 questions that span a range of contexts, including indoor and outdoor scenes, paintings, and sketches. This dataset is crafted to assess the capability of LVLMs in tackling more challenging tasks and their adaptability to new domains.[7]

## C.3 METRICS

**Metrics on the MME.** The evaluation dataset, $\mathcal{D}$ of the MME bechmark consists of two questions, $\{q_1, q_2\}$ regarding the same visual input, $\mathcal{V}$. Every question in $\mathcal{D}$ is a discriminating question. Based on the answers ("Yes" or "No") provided by the LVLMs, we can calculate the accuracy ($ACC$) for any $i$ as follows:

$$ACC(\mathcal{V}, q_i) = \begin{cases} 1 & \text{if LVLMs}(\mathcal{V}, q_i) = \text{Answer}(\mathcal{V}, q_i), \\ 0 & \text{otherwise.} \end{cases} \tag{12}$$

$ACC$, which is calculated for each query corresponding to an individual image, $ACC+$ (Fu et al., 2024) is calculated only when both queries associated with a single image are answered correctly. This metric is defined as follows:

$$ACC+(\mathcal{V}) = \begin{cases} 1 & \text{if LVLMs}(\mathcal{V}, q_i) = \text{Answer}(\mathcal{V}, q_i) \text{ for any } i, \\ 0 & \text{otherwise.} \end{cases} \tag{13}$$

MME score for each evaluated category is the summation of $ACC$ and $ACC+$.

**Metrics on the generative task.** Considering $R$ as the response by LVLMs for visual input, $V$, the following metrics can be delineated.

**CHAIR** (Rohrbach et al., 2018; Wang et al., 2023b) The *CHAIR* evaluates the occurrence of hallucinatory objects in responses to LVLMs. *CHAIR* uses an annotated list of objects $A=\{a_{obj}^1, a_{obj}^2, \ldots, a_{obj}^n\}$ to calculate how often hallucinated objects appear in the responses. Let $R=\{r_{obj}^1, r_{obj}^2, \ldots, r_{obj}^m\}$ be the list of objects mentioned in the response of LVLMs, the formula for *CHAIR* is given as:

$$CHAIR = 1 - \frac{len(R \cap A)}{len(R)}. \tag{14}$$

**Cover** (Wang et al., 2023b) The *Cover* metric measures how completely the objects in the response cover the identified objects in the image. *Cover* calculates the ratio of objects mentioned in the response to the total objects listed. The formula for *Cover* is:

$$Cover = \frac{len(R \cap A)}{len(A)}. \tag{15}$$

**Hal** (Wang et al., 2023b) The *Hal* metric quantifies the presence of hallucinations by checking if the *CHAIR* value is not zero, indicating the presence of hallucinations. The *Hal* is presented by the following formula:

$$Hal = \begin{cases} 1 & \text{if } CHAIR \neq 0, \\ 0 & \text{otherwise.} \end{cases} \tag{16}$$

**Cog** (Wang et al., 2023b) The *Cog* metric evaluates whether the hallucinations in LVLMs responses resemble human cognition. The *Cog* calculates the ratio of the human hallucinatory object targets, denoted as $H=\{h_{obj}^1, h_{obj}^2, \ldots, h_{obj}^n\}$ to the objects mentioned in the response. The formula for *Cog* is:

$$Cog = \frac{len(R \cap H)}{len(R)}. \tag{17}$$

---

[6]https://github.com/junyangwang0410/AMBER.git

[7]https://huggingface.co/datasets/liuhaotian/llava-bench-in-the-wild

***AMBER Score*** ([Wang et al., 2023b](#)) The *AMBER Score* metric evaluates the comprehensive performance of LVLMs for generative tasks and discriminative tasks. This score combines the CHAIR metric for generative tasks with the F1 metric for discriminative tasks. The formula representing the AMBER Score is as follows:

$$AMBER\ Score = \frac{1}{2} \times (1 - CHAIR + F1). \tag{18}$$

# D ADDITIONAL EXPERIMENTS

## D.1 INFERENCE TIME AND OPERA

Table 6: Comparison of inference time and performance on POPE-COCO-Random benchmark.

| | | | | LLaVA-1.5 | |
|---|---|---|---|---|---|
| **Method** | **Acc.** | **Prec.** | **Rec.** | **F1** | **tokens/sec** |
| *base* | 84.47 | 83.35 | 86.13 | 84.72 | 24.44 |
| VCD | 84.80 | 83.00 | 87.53 | 85.20 | 11.53 |
| M3ID | 86.00 | 85.11 | 87.27 | 86.18 | 13.14 |
| **AvisC** | 87.93 | 88.24 | 87.53 | 87.88 | 12.28 |
| OPERA (Beam=2) | 89.35 | 90.37 | 88.80 | 89.58 | 0.17 |

Tab. 6 presents an efficiency and performance comparison between contrastive decoding methods (AvisC, M3ID, OPERA, and VCD) and AVISC. Inference speed is measured with a TiTAN RTX GPU on the POPE-COCO-Random benchmark. OPERA introduces the concept of an "anchor token" and uses this token to guide sentence generation and rollback, thereby mitigating hallucinations. OPERA is implemented on the beam search decoding method of LLMs, so a fair comparison with AvisC is not possible. However, OPERA showed the best performance overall. However, its inference speed was approximately x72.23 slower than AVISC.

## D.2 ALTERNATIVES TO ZERO-OUT

Table 7 presents the results of ablation experiments on various deactivation schemes for non-blind image tokens using InstructBLIP ([Dai et al., 2024](#)) and LLaVA 1.5 ([Liu et al., 2023c](#)) models, evaluated on the POPE-COCO-random benchmark ([Li et al., 2023c](#)). We compare Zeros, Ones, Noise, and Mask. For InstructBLIP, Mask achieves the highest Accuracy and F1 score, while Zeros excels in Precision. Ones shows the highest Recall, and Noise provides balanced performance with high Precision and Recall. For LLaVA 1.5, Noise achieves the highest Accuracy and Precision, while Zeros shows balanced performance across all metrics. On average, using Zeros was the most effective in improving model performance by calibrating attention to image tokens.

Table 7: **Design choices for non-blind image token deactivation.**

| | **Case** | **Acc.** ↑ | **Prec.** ↑ | **Rec.** ↑ | **F1** ↑ |
|---|---|---|---|---|---|
| **InstructBLIP** | Zeros | 88.50 | 93.00 | 83.27 | 87.86 |
| | Ones | 82.50 | 75.48 | 96.27 | 84.62 |
| | Noise | 86.77 | 84.71 | 89.73 | 87.15 |
| | Mask | 88.53 | 90.14 | 86.53 | 88.30 |
| **LLaVA 1.5** | Zeros | 87.87 | 88.12 | 87.53 | 87.83 |
| | Ones | 79.97 | 72.22 | 97.40 | 82.94 |
| | Noise | 88.47 | 93.19 | 83.00 | 87.80 |
| | Mask | 84.77 | 86.29 | 82.67 | 84.44 |

## D.3 RESULTS OF LARGER LVLM

Tab. 8 presents the performance of each method on the POPE benchmark using the COCO dataset based on the LLaVA-1.5v-13B model. In this experiment setup, compared to the 7B small model shown in Tab. 1, the performance improvement of AVISC is even more pronounced. For other methods (*i.e.*, VCD, M3ID), the performance increase is slight or, in some cases, decreases depending on the metric. However, AVISC demonstrates robust performance improvement, remaining resilient to changes in the size of LVLMs.

Table 8: **Results of 13B models on COCO dataset.**

| Setup | Method | LLaVA-1.5 (13B) | | | |
|---|---|---|---|---|---|
| | | Acc. | Prec. | Rec. | F1 |
| Random | *base* | 83.17 | 79.49 | 89.40 | 84.15 |
| | VCD | 82.97 | 78.90 | 90.00 | 84.09 |
| | M3ID | 83.43 | 79.31 | 90.47 | 84.52 |
| | AVISC | 88.40 | 86.05 | 91.67 | 88.77 |
| Popular | *base* | 80.93 | 76.45 | 89.40 | 82.42 |
| | VCD | 79.67 | 74.59 | 90.00 | 81.57 |
| | M3ID | 80.90 | 75.94 | 90.47 | 82.57 |
| | AVISC | 85.73 | 81.94 | 91.67 | 86.53 |
| Adversarial | *base* | 76.03 | 70.74 | 88.80 | 78.75 |
| | VCD | 75.57 | 69.86 | 89.93 | 78.64 |
| | M3ID | 75.80 | 69.97 | 90.40 | 78.88 |
| | AVISC | 79.27 | 73.65 | 91.13 | 81.47 |

### D.4 POPE (LI ET AL., 2023C) WITH SINGLE-WORD CONSTRAINT

As shown in Tab. 9, we see that imposing a one-word response constraint on LVLMs leads to notable changes in performance compared to Tab. 1. Despite the change in query setup, AVISC shows the best performance on the POPE benchmark. Specifically, precision and recall vary significantly in the COCO random setup comparing scenarios with and without the instruction, "Please answer this question with one word." To mitigate these impacts and better evaluate discriminative capabilities, we designed experiments that allow the LVLMs to freely make judgments and provide explanations for these judgments rather than restricting them to answers in one word.

### D.5 DETAILED RESULTS ON MME-FULLSET

The detailed results on MME-Fullset are provided in Tab. 10. AVISC demonstrates substantial improvements in both LLaVA-1.5 and InstructBLIP across a wide range of perception and recognition tasks. These findings highlight the capability of AVISC to effectively handle diverse tasks, extending beyond hallucination mitigation, and suggest its potential to enhance the ability of LVLMs to accurately interpret and analyze visual information and query text appropriately.

### D.6 DETAILED RESULTS ON AMBER DISCRIMINATIVE TASKS

Tab. 11 presents the performance of the discriminative task on the AMBER benchmark across different categories. The discriminative task in the AMBER benchmark is divided into six categories: 'Existence', 'Attribute', 'State', 'Number', 'Action', and 'Relation', to evaluate the model's performance. For most categories, except for a few, both the LLaVA-1.5 and InstructBLIP models show performance improvements due to the applied AVISC.

### D.7 ADDITIONAL QUALITATIVE RESULTS

We provide additional qualitative results on all benchmarks (POPE (Li et al., 2023c), MME (Fu et al., 2024), AMBER (Wang et al., 2023b), and LLaVA-Bench (Liu et al., 2023c)) in Figs. 13 to 16. These highlight the differences between sentences generated by standard decoding (Base), VCD (Leng et al., 2023), and those produced by AVISC. The results demonstrate the effectiveness of AVISC in dealing with a variety of challenging visual contexts. Base and VCD often generate descriptions that include errors or hallucinations where elements not present in the image are described. In contrast, AVISC helps counteract these hallucinations, generating sentences that reflect a more accurate comprehension of the image.

Table 9: **POPE (Li et al., 2023c) results with one-word constraint.** We use the instruction "Please answer in one word." at the end of the query text.

| Setup | | Method | InstructBLIP (Dai et al., 2024) | | | | LLaVA 1.5 (Liu et al., 2023c) | | | |
|---|---|---|---|---|---|---|---|---|---|---|
| | | | Acc. | Prec. | Rec. | F1 | Acc. | Prec. | Rec. | F1 |
| MS-COCO | Random | *base* | 81.53 | 82.71 | 79.73 | 81.19 | 83.77 | 92.31 | 73.67 | 81.94 |
| | | VCD | 82.03 | 83.77 | 79.47 | 81.56 | 85.43 | 93.25 | 76.40 | 83.99 |
| | | AVISC | 86.03 | 95.53 | 75.60 | 84.41 | 84.67 | 97.88 | 70.87 | 82.21 |
| | Popular | *base* | 78.47 | 77.73 | 79.80 | 78.75 | 82.57 | 89.62 | 73.67 | 80.86 |
| | | VCD | 79.13 | 78.94 | 79.47 | 79.20 | 83.17 | 88.36 | 76.40 | 81.94 |
| | | AVISC | 84.27 | 91.45 | 75.60 | 82.77 | 83.67 | 95.25 | 70.87 | 81.27 |
| | Adversarial | *base* | 77.43 | 76.09 | 80.00 | 78.00 | 79.77 | 83.85 | 73.73 | 78.47 |
| | | VCD | 77.23 | 76.10 | 79.40 | 77.72 | 80.27 | 82.76 | 76.47 | 79.49 |
| | | AVISC | 81.83 | 86.20 | 75.80 | 80.67 | 81.83 | 90.99 | 70.67 | 79.55 |
| A-OKVQA | Random | *base* | 81.33 | 78.52 | 86.27 | 82.21 | 84.93 | 89.16 | 79.53 | 84.07 |
| | | VCD | 81.57 | 78.78 | 86.40 | 82.42 | 85.53 | 87.64 | 82.73 | 85.12 |
| | | AVISC | 87.10 | 89.95 | 83.53 | 86.62 | 87.33 | 95.09 | 78.73 | 86.14 |
| | Popular | *base* | 76.87 | 72.69 | 86.07 | 78.82 | 80.90 | 81.77 | 79.53 | 80.64 |
| | | VCD | 77.30 | 73.10 | 86.40 | 79.19 | 81.17 | 80.22 | 82.73 | 81.46 |
| | | AVISC | 82.47 | 81.79 | 83.53 | 82.65 | 85.03 | 90.08 | 78.73 | 84.03 |
| | Adversarial | *base* | 71.40 | 66.67 | 85.60 | 74.96 | 74.80 | 72.63 | 79.60 | 75.95 |
| | | VCD | 72.47 | 67.39 | 87.07 | 75.97 | 75.03 | 71.87 | 82.27 | 76.72 |
| | | AVISC | 76.47 | 73.16 | 83.60 | 78.03 | 79.27 | 79.58 | 78.73 | 79.16 |
| GQA | Random | *base* | 80.57 | 77.47 | 86.20 | 81.60 | 84.80 | 87.88 | 80.73 | 84.16 |
| | | VCD | 81.73 | 79.02 | 86.40 | 82.55 | 85.63 | 86.89 | 83.93 | 85.38 |
| | | AVISC | 85.30 | 88.57 | 81.07 | 84.65 | 87.40 | 95.17 | 78.80 | 86.21 |
| | Popular | *base* | 74.67 | 70.17 | 85.80 | 77.20 | 79.37 | 78.59 | 80.73 | 79.64 |
| | | VCD | 74.63 | 69.94 | 86.40 | 77.30 | 78.73 | 76.03 | 83.93 | 79.78 |
| | | AVISC | 80.63 | 80.37 | 81.07 | 80.72 | 83.33 | 86.66 | 78.80 | 82.54 |
| | Adversarial | *base* | 72.63 | 67.78 | 86.27 | 75.92 | 76.00 | 74.13 | 79.87 | 76.89 |
| | | VCD | 71.93 | 67.21 | 85.67 | 75.32 | 76.40 | 72.76 | 84.40 | 78.15 |
| | | AVISC | 77.60 | 75.91 | 80.87 | 78.31 | 80.37 | 81.52 | 78.53 | 80.00 |

# E    LICENSE OF ASSETS.

POPE (Li et al., 2023c) is made available under the MIT License. AMBER (Wang et al., 2023b) and LLaVA-Bench (Liu et al., 2023c) is available under Apache-2.0 License. InstructBLIP (Dai et al., 2024) is under BSD-3-Clause License and LLaVA (Liu et al., 2023c) is licensed under the Apache-2.0 License.

# F    BROADER IMPACTS

The release of our proposed AVISC for alleviating hallucinations in LVLMs comes with a wide range of positive and negative impacts.

**Positive impacts.** By mitigating hallucination, LVLMs can become more accurate and reliable tools for a wide range of applications, such as machine translation, chatbot development, and news generation.

**Negative impacts.** Our approach, AVISC, aimed at reducing hallucination, could heighten computational requirements, potentially resulting in higher expenses and greater energy use.

Table 10: **Results on MME-Fullset (Fu et al., 2024).**

| Task | Category | LLaVA 1.5 (Liu et al., 2023c) | | | | InstructBLIP (Dai et al., 2024) | | | |
|---|---|---|---|---|---|---|---|---|---|
| | | *base* | VCD | M3ID | AVISC | *base* | VCD | M3ID | AVISC |
| Perception | Existence | 173.57 (±8.16) | 172.14 (±8.09) | 178.33 (±6.83) | 189.29 (±1.89) | 170.19 (±11.12) | 172.62 (±8.92) | 173.89 (±10.52) | 184.76 (±5.56) |
| | Count | 110.00 (±15.82) | 117.14 (±8.76) | 107.22 (±14.78) | 104.76 (±11.66) | 89.52 (±11.04) | 98.33 (±15.99) | 89.72 (±13.44) | 82.85 (±12.16) |
| | Position | 100.47 (±18.78) | 103.33 (±20.56) | 96.39 (±5.52) | 106.19 (±13.93) | 67.62 (±14.04) | 71.90 (±13.42) | 72.72 (±14.77) | 74.76 (±6.19) |
| | Color | 125.24 (±15.91) | 119.52 (±8.58) | 127.50 (±8.28) | 127.86 (±9.13) | 114.76 (±9.60) | 117.14 (±10.70) | 110.56 (±7.20) | 131.43 (±4.76) |
| | Posters | 132.31 (±6.73) | 135.54 (±3.61) | 132.82 (±7.94) | 150.85 (±6.49) | 114.97 (±6.25) | 129.08 (±6.97) | 114.46 (±6.97) | 145.92 (±2.41) |
| | Celebrity | 114.56 (±6.45) | 118.09 (±7.69) | 113.38 (±0.21) | 125.59 (±2.50) | 113.38 (±3.95) | 123.82 (±4.99) | 114.12 (±2.91) | 120.29 (±7.90) |
| | Scene | 149.13 (±0.53) | 150.00 (±3.54) | 156.63 (±1.59) | 162.00 (±1.06) | 140.50 (±0.71) | 136.50 (±10.25) | 141.00 (±1.06) | 150.38 (±3.36) |
| | Landmark | 138.25 (±4.95) | 140.75 (±4.95) | 135.13 (±4.77) | 142.38 (±0.53) | 98.50 (±0.35) | 110.75 (±4.24) | 103.25 (±6.72) | 99.25 (±0.35) |
| | Artwork | 97.50 (±2.83) | 95.25 (±4.24) | 89.38 (±3.36) | 101.00 (±7.42) | 110.38 (±4.42) | 113.00 (±3.54) | 110.13 (±6.89) | 123.38 (±2.30) |
| | OCR | 91.25 (±19.45) | 101.25 (±1.77) | 96.25 (±15.91) | 143.75 (±5.3) | 87.50 (±21.21) | 91.25 (±8.84) | 85.00 (±10.61) | 68.75 (±5.3) |
| Recognition | Commonsense Reasoning | 100.36 (±2.53) | 96.79 (±5.56) | 87.14 (±12.12) | 102.86 (±7.07) | 96.43 (±1.01) | 107.14 (±8.08) | 99.64 (±2.53) | 101.79 (±6.57) |
| | Numerical Calculation | 80.00 (±7.07) | 66.25 (±8.84) | 76.25 (±12.37) | 65.00 (±14.14) | 68.75 (±1.77) | 66.25 (±15.91) | 71.25 (±22.98) | 73.75 (±5.30) |
| | Text Translation | 75.00 (±3.54) | 86.25 (±22.98) | 65.00 (±14.14) | 77.50 (±17.68) | 63.75 (±5.3) | 91.25 (±1.77) | 53.75 (±5.3) | 86.25 (±1.77) |
| | Code Reasoning | 62.50 (±10.61) | 61.25 (±1.77) | 71.25 (±15.91) | 71.25 (±5.30) | 73.75 (±5.30) | 57.50 (±0.00) | 81.25 (±1.77) | 76.25 (±5.3) |

Table 11: **Results on AMBER discriminative tasks (Wang et al., 2023b).**

| Category | LLaVA 1.5 (Liu et al., 2023c) | | | | InstructBLIP (Dai et al., 2024) | | | |
|---|---|---|---|---|---|---|---|---|
| | *base* | VCD | M3ID | AVISC | *base* | VCD | M3ID | AVISC |
| Existence | 68.55 (±0.21) | 67.15 (±1.91) | 68.50 (±0.14) | 75.35 (±0.21) | 72.05 (±0.49) | 73.20 (±1.27) | 72.95 (±0.21) | 81.35 (±0.07) |
| Attribute | 67.85 (±0.49) | 69.50 (±1.27) | 68.20 (±0.42) | 69.80 (±0.85) | 68.40 (±0.14) | 69.90 (±0.14) | 69.15 (±0.92) | 70.80 (±1.56) |
| State | 65.55 (±0.35) | 67.80 (±0.28) | 65.75 (±0.64) | 68.40 (±1.70) | 70.55 (±0.64) | 72.40 (±0.00) | 70.70 (±0.85) | 72.85 (±1.77) |
| Number | 69.05 (±0.78) | 68.50 (±2.40) | 68.95 (±0.92) | 67.10 (±1.84) | 60.90 (±0.00) | 60.70 (±0.85) | 61.80 (±0.71) | 60.85 (±0.49) |
| Action | 78.50 (±3.96) | 81.90 (±3.39) | 81.50 (±1.84) | 84.50 (±3.25) | 74.95 (±2.05) | 79.05 (±2.62) | 78.70 (±1.27) | 85.20 (±2.40) |
| Relation | 58.80 (±4.10) | 57.75 (±0.07) | 59.70 (±3.39) | 60.50 (±0.14) | 56.05 (±1.63) | 58.00 (±1.41) | 57.00 (±1.98) | 54.65 (±2.76) |

Overall, the potential positive impacts of research on reducing hallucination in LVLMs surpass the potential negative consequences. By addressing the hallucination problem, we can enhance the trustworthiness of LVLMs.

**InstructBLIP**

**Query**: Is there a chair in the image?

GT: No

**Base**: Yes, there is a wooden chair in the image, used primarily by the children.

**VCD**: There is a chair in the image, with kids sitting on it.

**M3ID**: Yes, there is a chiar in the image.

**AvisC**: There are several children sitting at tables on mats, with no visible chairs in the image.

**LLaVA-1.5**

**Query**: Is there a tv in the image?

GT: No

**Base**: Yes, there is a tv in the image. It sits atop a large modern dresser in the dining and living room area.

**VCD**: Yes, there is a flat screen television in the living room.

**M3ID**: Yes, there is a TV in the image, located in the living room area of the house.

**AvisC**: No, there is no TV in the image. The image features a dining room table with chairs, but it does not show any TV present in the image.

**InstructBLIP**

**Query**: Is there a chair in the image?

GT: Yes

**Base**: No, there isn't a chair in the image.

**VCD**: There is no chair in the image.

**M3ID**: No, there is no chair in the image.

**AvisC**: There is a step stool or a chair in the image, which the little girl is standing on while brushing her teeth.

**LLaVA-1.5**

**Query**: Is there a chair in the image?

GT: Yes

**Base**: No, there is no chair in the image. The image features a boy standing by a wall, surrounded by a bed with a comforter and a pillow.

**VCD**: No, there is no chair in the image. The scene takes place in a bedroom.

**M3ID**: No, there is no chair in the image. The image features a young boy standing on a bed with a flowered blanket.

**AvisC**: Yes, there is a chair in the image, located next to the boy standing on the pillows.

Figure 13: **Qualitative examples on POPE (Li et al., 2023c).**

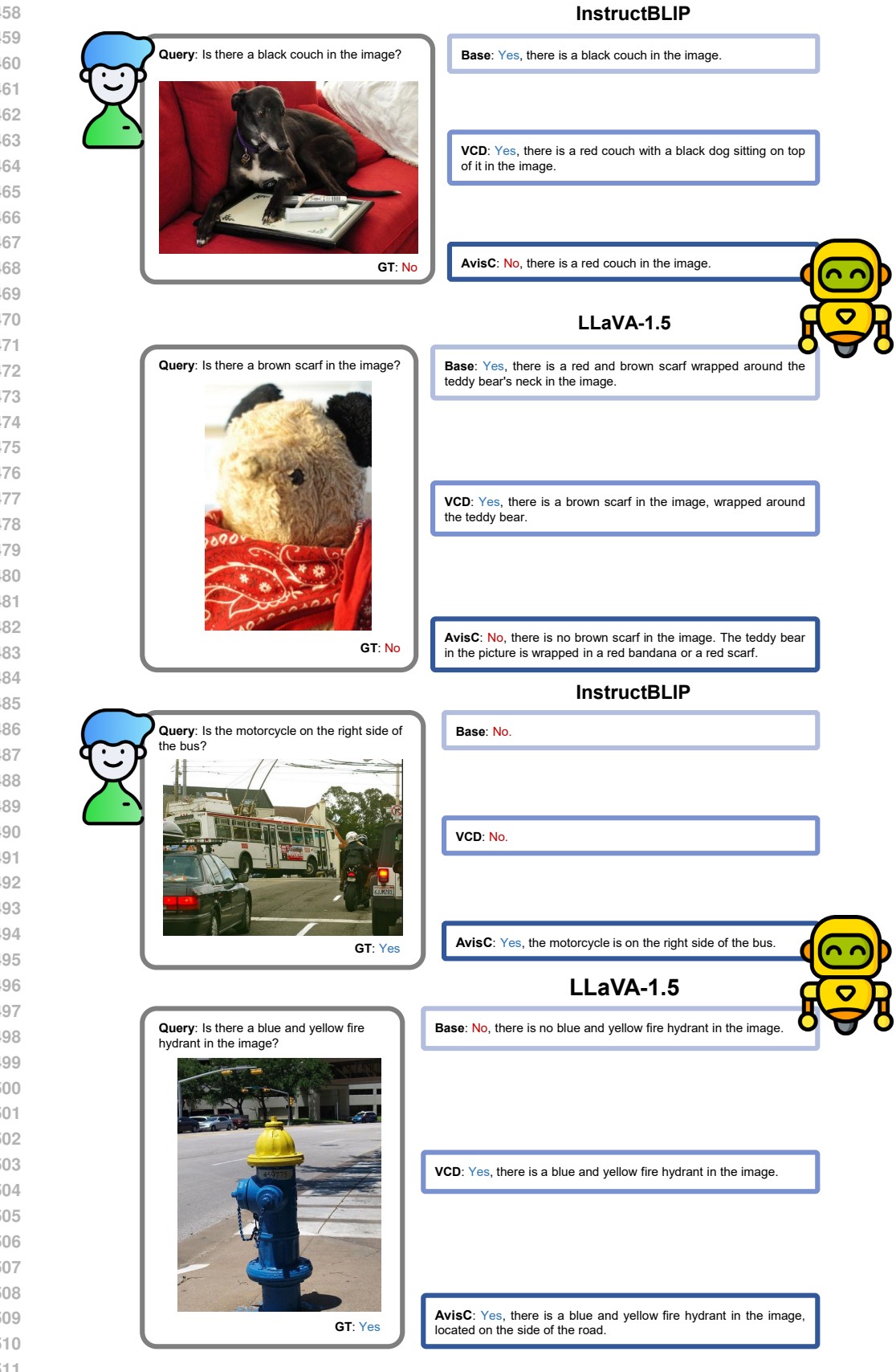

Figure 14: **Qualitative examples on MME (Fu et al., 2024).**

**InstructBLIP**

**Query**: Describe this image.

**Base**: In the video, a young girl is lying in a large ball pit, surrounded by various sized and colorful balls. She seems to be playing and enjoying herself, smiling and laughing as she plays with her new surroundings.

**VCD**: In the image, a young child is playing a ball pit, surrounded by colorful balls of different sizes and shapes. The child is fully immersed in the fun and enjoying the experience.

**AvisC**: The image features a little girl happily laying inside a ball pit filled with multiple colorful balls. She is having a good time as she smiles while being surrounded by the balls. The ball pit is quite full, with numerous balls placed across the area, indicating the child's enjoyment.

**Query**: Is there one surfboard in this image?

**GT**: Yes

**Base**: No, there are multiple surfboards in the image.

**VCD**: No, there are multiple surfboards in the image.

**AvisC**: Yes, there is only one surfboard in the image, being ridden by the man.

**InstructBLIP**

**Query**: Describe this image.

**Base**: The image displays a playful dog in the water, with a tennis ball in its mouth. The dog is floating in the water, seemingly enjoying its time playing in the water. The dog is a medium-sized breed, likely a retriever, and is centered in the water.

**VCD**: The image showcases a brown dog swimming in a large body of water, holding a toy in its mouth. The dog is covered by water as it moves through the water, possibly retrieving the toy or enjoying the activity of swimming.

**AvisC**: The image features a golden retriever swimming in a body of water, carrying a piece of something in his mouth. The dog is skillfully navigating the water while holding onto the object, likely a toy or something else it found during its swim.

**Query**: Are there two lamps in this image?

**GT**: No

**Base**: Yes, there are two lamps in this image. Specifically there are two white bedside table lamps on either side of the bed.

**VCD**: Yes, there are two lamps in the image. One is located on the bedside table and the other is on the nightstand

**AvisC**: No, there is only one lamp in the image, placed next to the bed on a side table.

Figure 15: **Qualitative examples of InstructBLIP (Dai et al., 2024) on AMBER (Wang et al., 2023b).**

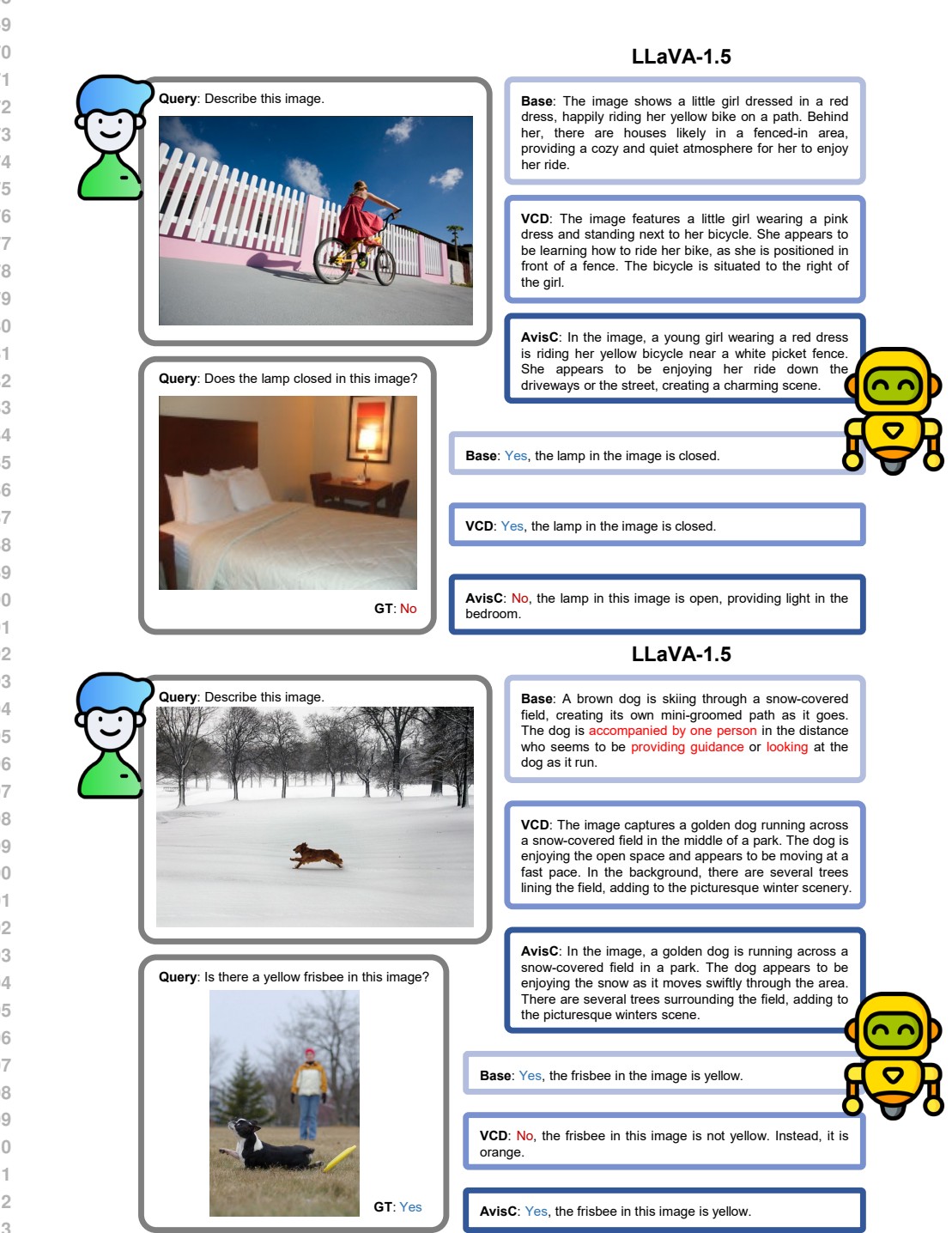

Figure 16: **Qualitative examples of LLaVA-1.5 (**Liu et al., 2023b**) on AMBER (**Wang et al., 2023b**).**

**Query**: Describe this photo in detail.

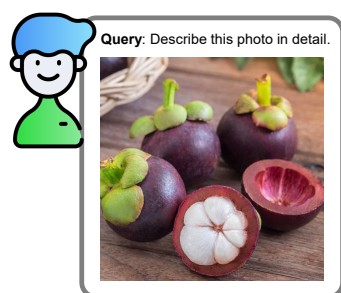

**Base**: Initially, this image portrays three green fruits, possibly ripe figs, with a white spot, harmlessly covering the inside of the fruit. They nestle among leaves, waiting for consumption or simply displayed for viewing. In addition to the three apples, their juxtaposition creates an aesthetic scene. The overall frame is filled with an inviting display of fresh fruits for those who appreciate or consume these delicious treats.

**VCD**: The image showcases a table with several fresh, ripe fruits cut in half. These fruits are not apples but rather a unique type of fruit that has a slightly purple tint, and their appearance is similar to grapefruits. The table displays four cut pieces of this fruit.

**AvisC**: The image features four plump purple fruit, possibly plums, placed on a wooden table. Three of the fruits are still whole, while the fourth fruit appears to be split in half, revealing its interior. The table provides an ideal surface for these fruits to be displayed on.

**Query**: Describe this photo in detail.

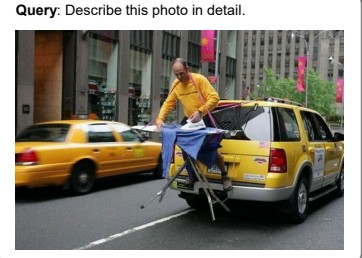

**Base**: In this bustling urban scene, a man in a yellow shirt is perched atop the back of a taxi or a yellow vehicle, possibly performing maintenance work on the car. This unique scene could incorporate elements from promoting an open source project such as a website.

**VCD**: The image depicts a busy city street with various vehicles and people. A man is prominently riding on the back of a yellow car, holding a blue shirt over the open hood of the vehicle. The car is parked on the side of the road.

**AvisC**: The image features a busy city street where a man is ironing clothes while sitting on the back of a yellow taxi cab. The taxi is parked, allowing the man to attend to his laundry amidst the urban bustle.

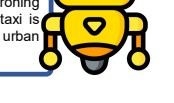

Figure 17: **Response comparison on LLaVA-Bench (Liu et al., 2023c).** Hallucinations are colored in red. AVISC demonstrates a robust understanding of images and reduces hallucinations in responses.

