# OpenReview forum: "Don't Miss the Forest for the Trees: Attentional Vision Calibration for Large Vision Language Models"
_ICLR.cc/2025/Conference — ICLR 2025 Conference Withdrawn Submission_

### Official Review · Reviewer_4Ged · 2024-10-16

**Soundness:** 3
**Presentation:** 3
**Contribution:** 2
**Rating:** 5
**Confidence:** 4

**Summary:**

The paper presents AVISC, a method to reduce hallucinations in LVLMs by adjusting attention to key visual tokens. It's lightweight, requires no extra training, and shows consistent improvement across benchmarks. However, I still have some concerns about this article, specifically in regard to the weaknesses and questions.
If these concerns are addressed, I will consider raising my score.

**Strengths:**

- The paper introduces a new method, AVISC, that helps reduce hallucinations in LVLMs by adjusting attention to focus on important visual tokens. This approach is simple yet effective.

- AVISC is a lightweight, post-hoc method that works without needing extra training or additional models, making it easy to apply across different LVLMs.

- The method shows consistent improvement in performance on standard hallucination benchmarks

**Weaknesses:**

- The authors' analysis of attention raises some concerns for me, as it seems somewhat coarse-grained, and the way image attention functions across layers appears to vary. Additionally, the occurrence of blind tokens might be better explained by attention sinks[1], which may not significantly affect the model's overall performance.

- I am confused by the experimental results about POPE in Tables 1 and 9, as they do not seem to fully align with the result from LLAVA.

- The authors did not perform more extensive evaluations on comprehensive and VQA benchmarks such as MMbench and VQAv2, which are crucial for assessing the model's overall performance.


[1] Efficient Streaming Language Models with Attention Sinks

**Questions:**

- Can the author provide a more fine-grained attention analysis, such as by layer and head, and analyze the correlation between blind token and the performance on other benchmarks?
- Could the authors clarify the specific settings followed in the experiments presented in Table 1 and 9? Additionally, how do these settings differ from those used in LLaVA?
- How does AVISC perform on comprehensive and VQA benchmarks such as MMbench and VQA?
- Does AVISC negatively affect other abilities of VLLMs, such as VQA and reasoning? This is especially relevant considering that many previous contrastive decoding methods exhibit similar shortcomings.
- Is this decoding method useful in more advanced VLLMs, such as Intern-VL, VILA, etc.?

---

### Official Review · Reviewer_ZhNL · 2024-10-17

**Soundness:** 3
**Presentation:** 2
**Contribution:** 3
**Rating:** 3
**Confidence:** 3

**Summary:**

The authors first review the attention pattern in transformer-based multimodal large models and discover that the models' focus is not aligned with the objects. Based on the above, this paper proposed a contrastive decoding method for Lagre vision language models, termed AVISC. During the generation process, AVISC retains those tokens that receive higher attention and correspond to meaningless areas as a reference for default decoding. The experiments show the performance improvement

**Strengths:**

1. The proposed AVISC methd is easy to follow and obviously improve the performance.

**Weaknesses:**

1. The writing of the article needs to be improved.
2. The proposed approach is not sufficiently motivated and leak a necessary baseline.

**Questions:**

1. The author needs to add a comparison with random zero-out of the same number of tokens.
(a) At the motivation level, when zero-out > $\mu$+$\sigma$, significantly more tokens will be zero-out, resulting in greater performance loss, and it is unconvincing to prove that zero-out blind-token has little effect on the original prediction logits.
(b) At the performance level, constructing biased prediction is intended to be compared with the original prediction, and the author needs to compare with random zero-out to illustrate the effectiveness of retaining blind tokens.

2. The authors need to explain the necessity and motivation of layer selection, and preferably provide ablation experiments with different P.

3. The font size in the picture is too small, such as Fig.4.

**Details Of Ethics Concerns:**

The authors must further demonstrate the relationship between their method and blind-token to better explain its motivation and the source of its performance.

---

### Official Review · Reviewer_uhN2 · 2024-11-02

**Soundness:** 3
**Presentation:** 3
**Contribution:** 2
**Rating:** 5
**Confidence:** 4

**Summary:**

This paper introduces a method called Attentional Vision Calibration (AVISC) to mitigate hallucinations in Large Vision Language Models (LVLMs). AVISC dynamically adjusts attention during the decoding phase to identify and reduce over-reliance on “blind tokens”. Key steps include layer selection, blind token identification, and contrastive decoding. Contrastive decoding leverages differences between the original and biased inputs to balance attention distribution, effectively decreasing the likelihood of hallucinations.

**Strengths:**

(1) AVISC operates directly during the decoding phase, requiring no additional training, auxiliary models, or complex self-feedback mechanisms, making it lightweight and easy to integrate into existing LVLMs.

(2) The authors’ discovery of the blind tokens phenomenon is very insightful. They also demonstrated this phenomenon through experiments and by visualizing bounding boxes and visual tokens.

**Weaknesses:**

(1) According to the experimental results, dynamically reducing attention dependence on irrelevant “blind tokens” is effective in reducing hallucinations. However, I am still concerned that this contrastive decoding approach may harm the original conversational abilities of LVLMs. The authors have only evaluated AVISC on hallucination benchmarks and have not tested it on more general benchmarks.

(2) Do other models, such as Qwen, also exhibit the same phenomenon, or more finely aligned models like InternVL?

**Questions:**

(1) Could the authors provide additional experiments on non-hallucination benchmarks, such as MME, MMbench, VQAv2, etc.? See Weaknesses (1) for specific reasons.

---

### Official Review · Reviewer_onHL · 2024-11-03

**Soundness:** 3
**Presentation:** 3
**Contribution:** 2
**Rating:** 5
**Confidence:** 4

**Summary:**

This paper identifies and addresses the "blind tokens" phenomenon in Large Vision Language Models (LVLMs), where models inappropriately focus attention on image tokens lacking meaningful semantic information. To address this issue, the authors propose Attentional Vision Calibration (AVISC), a decoding technique that identifies blind tokens and adjusts their influence through contrastive decoding, requiring no additional training or external models. Experimental results on multiple benchmarks (POPE, MME, and AMBER) demonstrate that AVISC consistently outperforms existing methods in reducing hallucinations and improving model accuracy.

**Strengths:**

Strengths:
1. The paper is well-written and organized.
2. The proposed AVISC method is straightforward to implement and requires no additional training or external models.

**Weaknesses:**

1. While the paper focuses on LLaVA-7B and InstructBLIP, the evaluation could have included more competitive models such as MiniCPM-V, LLaVA-13B, LLaVA-34B, and various grounding models like GLAMM.
2. The method requires computing logits twice - once with the original input and once with the biased input - essentially doubling the inference time. This computational overhead could be problematic in real-world applications where time efficiency is crucial. The paper would benefit from a more detailed discussion of this trade-off between performance improvement and computational cost.
3. The paper only compares AVISC with other decoding-based methods (VCD and M3ID), but misses comparisons with other important inference-time approaches for reducing hallucinations in LVLMs, such as OPERA[1].


[1] OPERA: Alleviating Hallucination in Multi-Modal Large Language Models via Over-Trust Penalty and Retrospection-Allocation. CVPR 2024

**Questions:**

1. The performance degradation in counting tasks is interesting. What is the authors' hypothesis for why AVISC performs worse specifically on counting tasks?
2. Are there certain types of visual information that may be encoded in blind tokens that are actually useful for specific tasks like counting?

---

### Note · Authors · 2024-11-14

**Comment:**

We sincerely appreciate the valuable reviews from the reviewers. We will reflect the comments and suggestions to improve the manuscript.

**Withdrawal Confirmation:**

I have read and agree with the venue's withdrawal policy on behalf of myself and my co-authors.